# Urban inland wintertime $N_2O_5$ and $ClNO_2$ influenced by snow-covered ground, air turbulence, and precipitation

Kathryn D. Kulju[1], Stephen M. McNamara[1], Qianjie Chen[1†], Hannah S. Kenagy[1], Jacinta Edebeli[1,2], Jose D. Fuentes[3], Steven B. Bertman[4], Kerri A. Pratt[1,5*]

[1]Department of Chemistry, University of Michigan, Ann Arbor, MI 48109, USA

[2]Paul Scherrer Institut, 5232 Villigen, Switzerland

[3]Department of Meteorology and Atmospheric Science, Pennsylvania State University, University Park, Pennsylvania 16802, USA

[4]Institute of the Environment and Sustainability, Western Michigan University, Kalamazoo, Michigan 49008, USA

[5]Department of Earth and Environmental Sciences, University of Michigan, Ann Arbor, MI 48109, USA

[†]Current: Department of Civil and Environmental Engineering, The Hong Kong Polytechnic University, Hong Kong SAR, China

*Correspondence to*: Kerri A. Pratt (prattka@umich.edu)

**Abstract.** The atmospheric multiphase reaction of dinitrogen pentoxide ($N_2O_5$) with chloride-containing aerosol particles produces nitryl chloride ($ClNO_2$), which has been observed across the globe. The photolysis of $ClNO_2$ produces chlorine radicals and nitrogen dioxide ($NO_2$), which alter pollutant fates and air quality. However, the effects of local meteorology on near-surface $ClNO_2$ production are not yet well understood, as most observational and modeling studies focus on periods of clear conditions. During a field campaign in Kalamazoo, Michigan from January-February 2018, $N_2O_5$ and $ClNO_2$ were measured using chemical ionization mass spectrometry, with simultaneous measurements of atmospheric particulate matter and meteorological parameters. We examine the impacts of atmospheric turbulence, precipitation (snow, rain) and fog, and ground cover (snow-covered and bare ground) on the abundances of $ClNO_2$ and $N_2O_5$. $N_2O_5$ mole ratios were lowest during periods of lower turbulence and were not

statistically significantly different between snow-covered and bare ground. In contrast, $ClNO_2$ mole ratios were highest, on average, over snow-covered ground, due to saline snowpack $ClNO_2$ production. Both $N_2O_5$ and $ClNO_2$ mole ratios were lowest, on average, during rainfall and fog because of scavenging, with $N_2O_5$ scavenging by fog droplets likely contributing to observed increased particulate nitrate concentrations. These observations, specifically those during active precipitation and with snow-covered ground, highlight important processes, including $N_2O_5$ and $ClNO_2$ wet scavenging, fog nitrate production, and snowpack $ClNO_2$ production, that govern the variability in observed atmospheric chlorine and nitrogen chemistry and are missed when considering only clear conditions.

## 1 Introduction

Atmospheric halogen radicals are highly oxidizing agents of tropospheric pollutants (Simpson et al., 2015). Following nighttime formation, the photolysis of nitryl chloride ($ClNO_2$) upon sunrise is a source of chlorine radicals (**R1**) at a time when other oxidants, including the hydroxyl radical (OH), are less abundant (Young et al., 2014), leading to enhanced oxidation of volatile organic compounds (Osthoff et al., 2008). $ClNO_2$ photolysis also releases $NO_2$ (**R1**), thus recycling nitrogen oxides ($NO_x=NO+NO_2$) that drive ozone formation (Crutzen, 1979).

$$\mathbf{ClNO_2 + h\nu \rightarrow Cl + NO_2} \qquad \text{R1}$$

$ClNO_2$ is formed by the multiphase reaction of dinitrogen pentoxide ($N_2O_5$) on a chloride-containing surface (**R2**), particularly sea spray aerosol (Finlayson-Pitts and Pitts, 1989; Osthoff et al., 2008).

$$\mathbf{N_2O_{5\,(g)} + Cl^-_{(aq)} \rightarrow\ ClNO_{2\,(g)} + NO^-_{3\,(aq)}} \qquad \text{R2}$$

In the Northern Hemisphere, surface-level $ClNO_2$ abundance is simulated to be highest during winter; this is thought to be due to greater $N_2O_5$ abundances, shallower mixed layer heights or even stable boundary layers, lower air temperatures, and higher $ClNO_2$ yields (Sarwar et al., 2014). $ClNO_2$ production has been previously studied in the laboratory following the reaction of $N_2O_5$ with aqueous aerosols (e.g.

Behnke et al., 1997; Bertram and Thornton, 2009; Roberts et al., 2009; Thornton and Abbatt, 2005) and frozen solutions (Lopez-Hilfiker et al., 2012). A recent modeling study suggests that $ClNO_2$ may be produced from heterogeneous reaction on the snowpack, in addition to aerosols (Wang et al., 2020). In addition to marine and coastal environments, $ClNO_2$ has been measured in inland environments, including Boulder, Colorado, USA (Riedel et al., 2013; Thornton et al., 2010), , Calgary, Alberta, Canada (Mielke et al., 2011), Frankfurt, Germany (Phillips et al., 2012), Ji'nan, Shandong, China (e.g. Wang et al., 2017), and southwest of Baoding, Hebei, China (e.g. Tham et al., 2018); in these inland environments, $ClNO_2$ abundance is typically hundreds of parts per trillion (ppt). Recently, a study in Ann Arbor, Michigan identified road salt aerosol as the dominant aerosol chloride source for $ClNO_2$ production during winter (McNamara et al., 2020). Measurements in Kalamazoo, Michigan also identified the road salt-contaminated snowpack as a $ClNO_2$ source (McNamara et al., 2021). A study in coastal British Columbia, Canada suggested scavenging of $ClNO_2$ by rain and/or fog droplets as a potential loss process (Osthoff et al., 2018). However, the authors pointed out that scavenging of the nitrate radical ($NO_3$), $N_2O_5$, and $ClNO_2$ have not been constrained by laboratory investigations (in contrast to other gases like sulfur dioxide ($SO_2$) and ammonia ($NH_3$)) and so periods of precipitation were excluded from subsequent calculations of $N_2O_5$ uptake and $ClNO_2$ yield (Osthoff et al., 2018).

$N_2O_5$, the precursor to $ClNO_2$, is formed from the reaction of $NO_2$ with $NO_3$ (**R3**), which is formed from the reaction of $NO_2$ with ozone ($O_3$, **R4**). The formation of $N_2O_5$ from $NO_2$ and $NO_3$ is a temperature-dependent equilibrium, with $N_2O_5$ production favored at lower temperatures (Asaf et al., 2010; Wagner et al., 2013). At a $NO_2$ background level of 1 parts per billion (ppb), the ratio of $N_2O_5$:$NO_3$ (**R3**) is ~1 at 295 K, but this $N_2O_5$:$NO_3$ ratio is ~10 at 278 K (Chang et al., 2011). Loss of $N_2O_5$ is an important terminal sink for nitrogen oxides ($NO_x = NO + NO_2$) in the troposphere (Simpson et al., 2015). Long-term data show that direct $N_2O_5$ loss via hydrolysis, to produce nitric acid ($HNO_3$, **R5**), is most important during winter, and indirect $N_2O_5$ loss (removal of $NO_3$ via reaction with hydrocarbons and NO, **R6-R7**) is most important during summer (Allan et al., 1999; Geyer et al., 2001; Heintz et al., 1996).

$$\textbf{NO}_3 + \textbf{NO}_2 + \textbf{M} \rightleftharpoons \textbf{N}_2\textbf{O}_5 + \textbf{M} \qquad \text{R3}$$

$$\textbf{O}_3 + \textbf{NO}_2 \longrightarrow \textbf{O}_2 + \textbf{NO}_3 \qquad \text{R4}$$

$$N_2O_{5\,(g)} + H_2O_{(l)} \longrightarrow HNO_{3\,(aq)} \qquad \text{R5}$$

$$R - C = C - R' + NO_3 \longrightarrow R - C - C(NO_3) - R' \qquad \text{R6}$$

$$NO + NO_3 \longrightarrow 2\,NO_2 \qquad \text{R7}$$

Experimental investigations of the impacts of meteorology on $N_2O_5$ abundance are primarily limited to observations of uptake by fog in coastal regions (Brown et al., 2016; Osthoff et al., 2006; Sommariva et al., 2009; Wood et al., 2005). In addition to forming $HNO_3$, hydrolysis of $N_2O_5$ can produce particle-phase nitrate ($NO_3^-$) (Brown et al., 2004; Osthoff et al., 2006). Particle-phase nitrate has been observed to increase, then subsequently decrease, during fog episodes, which is hypothesized to be the result of $N_2O_5$ hydrolysis to form nitrate, followed by wet removal of nitrate from the fog layer (Lillis et al., 1999).

The review by Chang et al. (2011) stated that future observation-based research is need to further investigate how $N_2O_5$ is affected by meteorological conditions, due to its impacts on $ClNO_2$ and particulate matter abundances, as well as on the oxidative capacity of the atmosphere. Many gaps remain in our understanding of the fates and production of $N_2O_5$ and $ClNO_2$, especially in inland locations, and how they are influenced by meteorological conditions such as precipitation events, fog, and turbulent mixing. Notably, Stanier et al. (2012) identified the impacts of fog and snow cover as important knowledge gaps in understanding wintertime atmospheric composition, and nitrate formation in particular, in the Midwest United States.

The SNow and Atmospheric Chemistry in Kalamazoo (SNACK) field campaign was conducted during January and February 2018 in Kalamazoo, MI on the campus of Western Michigan University (WMU). In our previous publication from this study, we showed photochemical snowpack HONO production due to snow nitrate photolysis (Chen et al., 2019). Through vertical gradient measurements on select nights of the SNACK field campaign, we showed that $N_2O_5$ deposits at the same rates over bare and snow-covered ground; whereas, while $ClNO_2$ deposits on bare ground, it can be emitted from the saline snow-covered ground, with snow chamber experiments confirming saline snow $ClNO_2$ production (McNamara et al., 2021). Here, we focus on the observational time series of near-surface $ClNO_2$ and its precursor $N_2O_5$ and examine the influences of precipitation (rain, snow) and fog, atmospheric turbulence,

ground cover (snow-covered vs bare ground), particulate chloride and nitrate, temperature, and relative humidity (RH) on the night-time abundances of these compounds, measured by chemical ionization mass spectrometry. This study provides new insights into the biases associated with modeling and observations focused on cloudless (clear) conditions, which has been shown to impact predictions of aerosol chemical composition (Christiansen et al., 2020).

## 2 Methods

The sampling site (42.28°N, 85.61°W) on the campus of WMU in Kalamazoo, MI was located next to a field and was approximately 90 m from a major roadway, as previously described by McNamara et al. (2021). As described below, measurements of trace gases ($N_2O_5$ and $ClNO_2$), $PM_{2.5}$ (particulate matter with a diameter $\leq 2.5$ μm) inorganic chemical composition, three-dimensional wind speed, and temperature were conducted at the field site from January 20 to February 24, 2018. Daily photographs and field notes were used to determine ground cover and spatial extent of snow cover.

Because $N_2O_5$ and $ClNO_2$ were present almost exclusively at night, we define "nocturnal"/"nighttime" as the period between 18:00 and 8:00 Eastern Standard Time (EST, Coordinated Universal Time (UTC)-5 h), which was approximately ±30 min from sunrise and sunset during the campaign. At the start of the campaign (January 20) sunrise was at 08:05 local time (eastern standard time, EST), and sunset was at 17:42. At the end of the campaign (February 24) sunrise was at 07:23 EST, and sunset was at 18:27.

## 2.1 Meteorological measurements

Air temperature and three-dimensional wind speed (u, v, and w) were measured from a height of 1.4 m and at a frequency of 20 Hz using a sonic anemometer (model CSAT3, Campbell Scientific Inc., Logan, UT). The sonic anemometer was not operational from February 20-21 due to complications associated with heavy rainfall. Friction velocity ($u^*$) was calculated from turbulent covariance of three-dimensional wind speed based on 30 minute averaging, where u', v', and w' are fluctuations about the 30

min mean wind speed in its zonal (u), meridional (v), and vertical (w) components, respectively (E1) (Stull, 1988).

$$u^* = (\overline{u'w'^2} + \overline{v'w'^2})^{\frac{1}{4}} \qquad \text{E1}$$

Kinematic heat flux ($w'T'$) was also calculated from sonic anemometer data, where w' and T' are deviations in vertical velocity and temperature from five-minute averages, respectively (Monin and Obukhov, 1954). Kinematic heat flux values were then further averaged to obtain 30 min time resolution quantities. This heat flux value describes the transport of thermal energy by eddies; negative values of w'T' indicate heat transport from the atmosphere to the surface and are associated with a temperature inversion (Stull, 1988).

Weather conditions (rain, snow, and fog) and pressure were recorded at the Kalamazoo−Battle Creek International Airport (KAZO), which was located ~7 km to the southeast; data were retrieved from Weather Underground (https://www.wunderground.com/history/daily/us/mi/kalamazoo/KAZO). Weather conditions were reported with a maximum time resolution of 1 h. This relatively long time resolution limits the use of higher frequency data from other measurements, and therefore, we use 30 min averaged data, with the assumption that the weather condition lasted the entire hour. Weather conditions were classified using reported National Weather Service designations: clear weather conditions include fair, cloudy, mostly cloudy, and partly cloudy; snowfall includes light snow, snow, heavy snow, and wintry mix; fog includes fog and haze; and rainfall refers to light rain, rain, heavy rain, and thunderstorms. Wind speed and temperature data were also obtained from this weather station to supplement the rain case study (February 20-21), during which data from the sonic anemometer were unavailable.

## 2.2 Chemical ionization mass spectrometry measurements

Measurements of $N_2O_5$ and $ClNO_2$ were conducted using a chemical ionization mass spectrometer (CIMS, THS Instruments) (Liao et al., 2011). The CIMS instrument uses iodide-water reagent ion clusters, $I(H_2O)^-$, to ionize analyte molecules, which are separated and quantified using a quadrupole mass analyzer. The CIMS was housed in a mobile laboratory trailer at the field site, and sampled ambient air at ~300 L min$^{-1}$ through a specialized inlet. The inlet was designed to prevent wall

losses of reactive species by allowing for the sampled air at the center of the ring to be de-coupled from the inlet walls (laminar flow), thereby avoiding wall surfaces (Huey et al., 2004; Neuman et al., 2002), as in previous campaigns (e.g., McNamara et al., 2019). The inlet consisted of a 30 cm long, 4.6 cm i.d. aluminum pipe attached to a stainless-steel ring torus 1.5 m above ground level. The airflow from this inlet was subsampled at 6.6 L min$^{-1}$ into a 48 cm long, 0.95 cm i.d. FEP Teflon tube and through a custom three-way heated valve (30°C) used to obtain calibration and background measurements. Of this airflow, an ozone monitor (model 205, 2B Technologies, Boulder, CO) sub-sampled 1.7 L min$^{-1}$, and 0.9 L min$^{-1}$ was sub-sampled into the CIMS ion-molecule reaction region, which was held at a constant pressure of 15.5 Torr. I(H$_2$O)$^-$ reagent ions (Slusher et al., 2004) were generated by passing iodomethane (CH$_3$I) in nitrogen (N$_2$) through a $^{210}$Po radioactive ion source. The ion-molecule reaction region was humidified using water vapor from an impinger to prevent changes in ambient RH from altering CIMS sensitivity (Kercher et al., 2009; McNamara et al., 2019).

CIMS background measurements were conducted for 2 min every 15 min by passing the ambient air flow through a scrubber containing glass wool and stainless-steel wool (heated to 120°C) which removed N$_2$O$_5$ and ClNO$_2$ with 96.4±0.8% and 89±1% efficiency (mean±95% confidence interval), respectively (McNamara et al., 2021). N$_2$O$_5$ was monitored at $m/z$ 235 (IN$_2$O$_5^-$), and ClNO$_2$ was monitored at $m/z$ 208 (I$^{35}$ClNO$_2^-$) and $m/z$ 210 (I$^{37}$ClNO$_2^-$), each with dwell times of 1.5 s. ClNO$_2$ was positively identified using its measured isotopic ratio (**Fig. S1**). The 3σ limits of detection (LOD), corresponding to the 2 min background periods, were 1.3 ppt and 0.4 ppt for N$_2$O$_5$ and ClNO$_2$, respectively. We report mole ratios as 30 min averages, for which the 3σ LODs for N$_2$O$_5$ and ClNO$_2$ are estimated to be 0.3 ppt and 0.1 ppt for N$_2$O$_5$ and ClNO$_2$, respectively, calculated in the same manner as Liao et al. (2011). CIMS measurement uncertainties, which include propagated uncertainties associated with calibrations and fluctuations in the background signal, are estimated as 22%+0.3 ppt and 22%+0.1 ppt for 30 min averaged N$_2$O$_5$ and ClNO$_2$ mole ratios, respectively. Calibrations in the field were conducted every 2 h by adding 0.2 L min$^{-1}$ of 12.3±0.2 ppb Cl$_2$ (in N$_2$) from a permeation source (VICI Metronics, Inc., Poulsbo, WA) to the ambient airflow. The permeation rate was measured by bubbling the permeation output into a solution of potassium iodide and measuring the oxidation product, triiodide (I$_3^-$), using UV-visible spectrophotometry at 352 nm (Liao et al., 2011). The instrument responses for N$_2$O$_5$ and ClNO$_2$ were

calibrated in the laboratory, with calibration factors relative to the response to $Cl_2$ obtained, as described in McNamara et al. (2019b).

$Cl_2$ was monitored as $I(Cl_2)^-$ at $m/z$ 197 and 199, each with dwell times of 0.5 s. The LOD for $Cl_2$ at $m/z$ 197 was 2.4 ppt (0.6 ppt for 30 min averaged data). $Cl_2$ was below its estimated LOD for 30 min averaging for 96% of the nighttime periods (and 91% of daytime periods), and therefore these limited data are not discussed. $HNO_3$ was also monitored as $I(HNO_3)^-$ at $m/z$ 190 with a dwell time of 0.5 s and calibrated offline relative to $Cl_2$ (McNamara et al., 2020). However, there was a high background signal due to poor scrubbing efficiency (12±1%), resulting in a high LOD of 43 ppt (11 ppt for 30 min averaged data). 40% of the nighttime $HNO_3$ data during the campaign were below the LOD estimated for 30 min averaging, and therefore these data are not discussed in detail in this work. These upper limits for $Cl_2$ and $HNO_3$ mole ratios are important to report, given limited measurements of these compounds in urban, snow-covered environments.

**2.3 Ambient ion monitor-ion chromatography (AIM-IC)**

$PM_{2.5}$ chloride ($Cl^-$) and nitrate ($NO_3^-$) were measured by an ambient ion monitor-ion chromatography instrument (AIM-IC; model 9000D, URG Corp., Chapel Hill, NC), as described in Chen et al. (2019). The AIM-IC and custom outdoor sampling inlet is described in detail by Markovic et al. (2012). Briefly, ambient air was sampled at 3 L min$^{-1}$ through a 2.5 μm cyclone at a height of 1.8 m. A parallel-plate wet denuder (PPWD) supplied with diluted $H_2O_2$ separated soluble inorganic trace gases. Particles entered a supersaturation chamber (SSC), where hygroscopic growth was initiated prior to an inertial particle separator. The PPWD and SSC were placed outside in an insulated and heated aluminum case to reduce the sampling line length. Trace gas and particle samples were collected every hour using concentrator columns (anion, UTAC-ULP1, ultra-trace anion concentrator ultralow pressure; cation, TCC-ULP1, trace cation concentrator ultralow pressure; Thermo Fisher Scientific, Waltham, MA) for measurements every 2-4 h (3 h after January 24) by an ion chromatograph (ICS-2100; Dionex Inc., Sunnyvale, CA). LiF was used as an internal standard. The 3σ LODs for $Cl^-$ and $NO_3^-$ were 0.004 and 0.05 μg m$^{-3}$, respectively, for 3 h sampling.

## 2.4 Aerosol size distribution measurements

Aerosol size distributions were measured using a scanning mobility particle sizer (SMPS, model 3082, TSI, Inc., Shoreview, MN), which measured electrical mobility diameter from 14.1-736.5 nm, and an aerodynamic particle sizer (APS, model 3321, TSI, Inc., Shoreview, MN), which measured aerodynamic diameter from 0.5-20 μm. The air was sampled through a 2.5 μm cyclone (URG Corp., Chapel Hill, NC) from an inlet height of ~3 m. This flow was split from a manifold with a total flow rate of 16.8 L min$^{-1}$ into foam-insulated copper tubing for each instrument; the SMPS and APS sub-sampled at 0.3 L min$^{-1}$ and 4.9 L min$^{-1}$, respectively.

## 3 Results and Discussion

The field campaign nights from January 20-February 24 were divided into categories to investigate the impacts of weather events (rain, snowfall, fog), ground cover (snow-covered and bare ground), and atmospheric turbulence on the near-surface (~1.5 m above ground) abundances of $N_2O_5$ and $ClNO_2$ (**Fig. 1**). Time periods that were below LOD (0.3 ppt and 0.1 ppt for 30 min averaged $N_2O_5$ and $ClNO_2$, respectively) are included in calculations as $0.5 \times LOD$. Data after 08:00 (approximately ±30 min from sunrise, which was at 08:07 on Jan 20 and 07:25 on Feb 24) are not included such that air entrainment from the residual boundary layer, discussed elsewhere (e.g. Tham et al., 2016), does not influence the results discussed below.

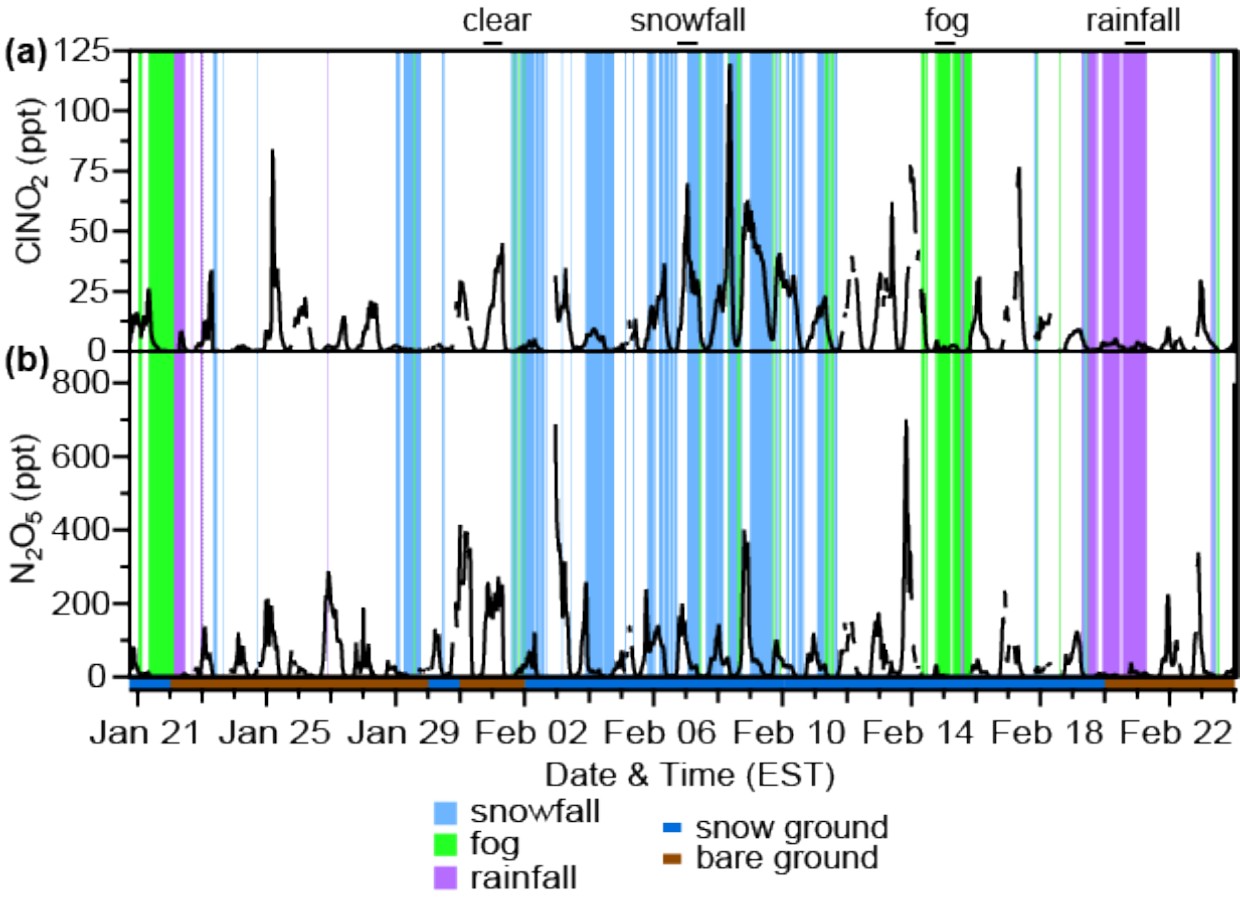

**Figure 1**: Mole ratios of 30 min averaged **(a)** $ClNO_2$ and **(b)** $N_2O_5$ during the campaign, and occurrence of snowfall (*light blue*), fog (*green*), and rainfall *(purple)*. The shading below the x-axis represents ground cover – snow *(blue)* or bare ground *(brown)*. The black bars on the top of the plot show the selected case study nights for each weather event type. Between 18:00 and 08:00 h EST, where n=number of 30 min periods, the air was clear 72% of the time [n=726; 363 h], snowfall occurred 16% of the time [n=157; 78.5 h], rainfall occurred 6% of the time [n=63; 31.5 h], and fog occurred 6% of the time [n=58; 29 h]. The ground was snow-covered 57% of the study [20 d] and was bare for 43% of the study [15 d]. Figure S3 gives further details about the occurrence of weather events (rainfall, fog, snowfall) in relation to friction velocity and ground cover.

### 3.1 Effects of rain, snow, and fog (campaign-wide)

The nighttime abundances of $N_2O_5$ and $ClNO_2$ during rain, snowfall, and fog were all significantly different ($p<0.05$, six t-tests) from clear conditions (**Fig. 2**). Campaign-wide average nighttime (18:00-08:00) $N_2O_5$ and $ClNO_2$ mole ratios during clear conditions and each type of weather event are listed in

Table 1, with additional data ($PM_{2.5}$ $Cl^-$ and $NO_3^-$, temperature, relative humidity, and friction velocity) for these time periods provided in Table S1. Here we discuss observations during these weather events

across the entire campaign; example case studies are discussed in Section 3.2. The average nighttime $N_2O_5$ mole ratios (±95% confidence interval) were 84±5 ppt, 47±2 ppt, 14±2 ppt, and 7.1±0.6 ppt during clear, snowfall, rain, and fog conditions, respectively (**Fig. 2**). In comparison to clear conditions, average $N_2O_5$ mole ratios were 37±5 ppt (1.8 times), 70±5 ppt (6.0 times), and 77±5 ppt (12 times) lower during snowfall, rain, and fog, respectively. The decrease in $N_2O_5$ abundance during fog suggests $N_2O_5$ uptake

by fog droplets, and is consistent with previous observations (Brown et al., 2016; Osthoff et al., 2006; Sommariva et al., 2009; Wood et al., 2005). More recently, a study by Osthoff et al. (2018) noted decreased $ClNO_2$ abundance during drizzle/rain and fog during Jul.-Aug. in coastal British Columbia. However, clear conditions are generally the focus of previous $N_2O_5$ and $ClNO_2$ studies (Chang et al., 2011; Simpson et al., 2015).

**Table 1:** Mean (±95% confidence interval) mole ratios of $N_2O_5$ and $ClNO_2$, $PM_{2.5}$ $Cl^-$ and $NO_3^-$ concentrations, temperatures, and relative humidity during each type of weather event (clear, snow, fog, and rain) and ground cover (bare and snow-covered ground) measured across the entire campaign, between 18:00-08:00 EST. The numbers of 30 min periods (n) and percentages of nighttime periods classified as each weather condition are included in parenthesis. Note that bare and snow-covered ground co-existed with the weather conditions, as discussed in Section 3.4. 95% confidence intervals are reported to describe the variabilities in 30 min averaged values of the parameters for the various weather and ground cover conditions.

| Weather or Ground Cover Condition | $N_2O_5$ (ppt) | $ClNO_2$ (ppt) | $[Cl^-]$ ($\mu g\ m^{-3}$) | $[NO_3^-]$ ($\mu g\ m^{-3}$) | Temperature (K) | Relative Humidity (%) |
|---|---|---|---|---|---|---|
| Clear (n=726, 72%) | 84±5 | 11.8±0.7 | 0.257±0.007 | 0.95±0.04 | 270.8±0.3 | 75.0±0.5 |
| Snowfall (n=157, 16%) | 47±2 | 16.8±0.7 | 0.258±0.006 | 0.81±0.03 | 265.8±0.2 | 83.0±0.3 |
| Fog (n=58, 6%) | 7.1±0.6 | 5.0±0.6 | 0.456±0.008 | 1.38±0.04 | 276.7±0.2 | 93.7±0.3 |
| Rain (n=63, 6%) | 14±2 | 2.27±0.06 | 0.22±0.01 | 0.126±0.007 | 282.1±0.2 | 90.2±0.4 |

| | | | | | | |
|---|---|---|---|---|---|---|
| Snow-covered ground (20 d, 57%) | 70±5 | 14.9±0.8 | 0.30±0.01 | 1.03±0.04 | 268.4±0.3 | 80.0±0.5 |
| Bare ground (15 d, 43%) | 68±4 | 7.0±0.5 | 0.21±0.01 | 0.67±0.03 | 274.8±0.3 | 75.8±0.6 |

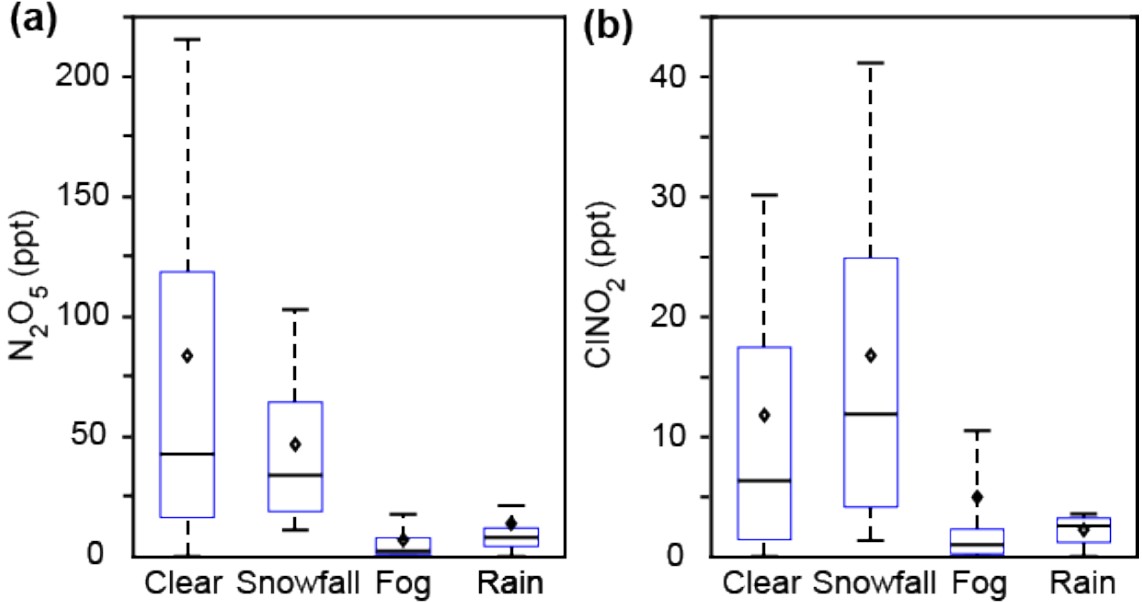

**Figure 2:** Box plots showing 30 min averaged mole ratios of (a) $N_2O_5$ and (b) $ClNO_2$ during clear conditions and weather events (snowfall, fog, and rain) from the entire campaign, January 20 - February 24. Bars represent the $10^{th}$, $50^{th}$, and $90^{th}$ percentiles, boxes represent the $25^{th}$ and $75^{th}$ percentiles, and diamonds represent the means. Only nighttime data between 18:00 and 08:00 EST are included. Data during all weather events (snowfall, fog, rain) are significantly different ($p < 0.05$, t-test) from clear conditions.

The average $ClNO_2$ mole ratios were 16.8±0.7 ppt during snowfall, 11.8±0.7 ppt in clear conditions, 5.0±0.6 ppt during fog, and 2.27±0.06 ppt when raining (Fig. 2). In comparison to clear conditions, average $ClNO_2$ mole ratios were 6.8±0.9 ppt (2.4 times) and 9.5±0.7 ppt (5.2 times) lower during fog and rain, respectively. Lower average abundances of $ClNO_2$ during fog and rainfall, compared to clear conditions, are consistent with previous observations (Osthoff et al., 2018) and were likely due

to scavenging either of $ClNO_2$ directly or its precursors (R2). In contrast, average $ClNO_2$ mole ratios were 5±1 ppt (1.4 times) higher during snowfall than clear conditions. This result is surprising, considering that its precursor, $N_2O_5$, showed lower mole ratios, on average, during snowfall in comparison to clear conditions. We hypothesize that snowpack $ClNO_2$ production contributes to this observation, which is discussed in *Sect. 3.3-3.4*. Particle-phase chloride and nitrate concentrations were not statistically significantly different between clear and snowfall conditions (p=0.96 and 0.08, respectively), nor were aerosol number or surface area concentrations (p=0.06 and 0.31, respectively), as discussed in *Sect. 3.5*. The effects of temperature and relative humidity are discussed in *Sect. 3.5*.

## 3.2 Effects of rain, snow, and fog (case study nights)

To further examine the behavior of $N_2O_5$ and $ClNO_2$ mole ratios in response to snowfall, rain, and fog, we present four nocturnal case study periods that were representative of the four different weather conditions (clear, snowfall, fog, and rain) observed during the campaign (**Fig. 3**). Case study nights were chosen to capture a sustained weather event (e.g. >7 h of clear conditions, snowfall, fog, or rainfall). Additionally, ground cover and friction velocity were matched as closely as possible for case study nights to the campaign-wide averages during different types of weather events. Additional data specific to the case studies is provided in the supplemental material (**Table S2**, **Fig. S4-S6**). The clear case night of Jan 31-Feb 01 had no precipitation or fog, an average u* of 0.16±0.01 m s$^{-1}$ (campaign average u* was 0.150±0.004 m s$^{-1}$ during nighttime clear conditions), and bare ground. $N_2O_5$ mole ratios were fairly stable around 200 ppt (average 200±16 ppt, range 75-274 ppt) throughout the night, with $ClNO_2$ mole ratios increasing steadily between 18:00-07:30 from 1.5 ppt to 45 ppt (average 23±5 ppt, range 0.6-4.5 ppt) (**Fig. 3a**).

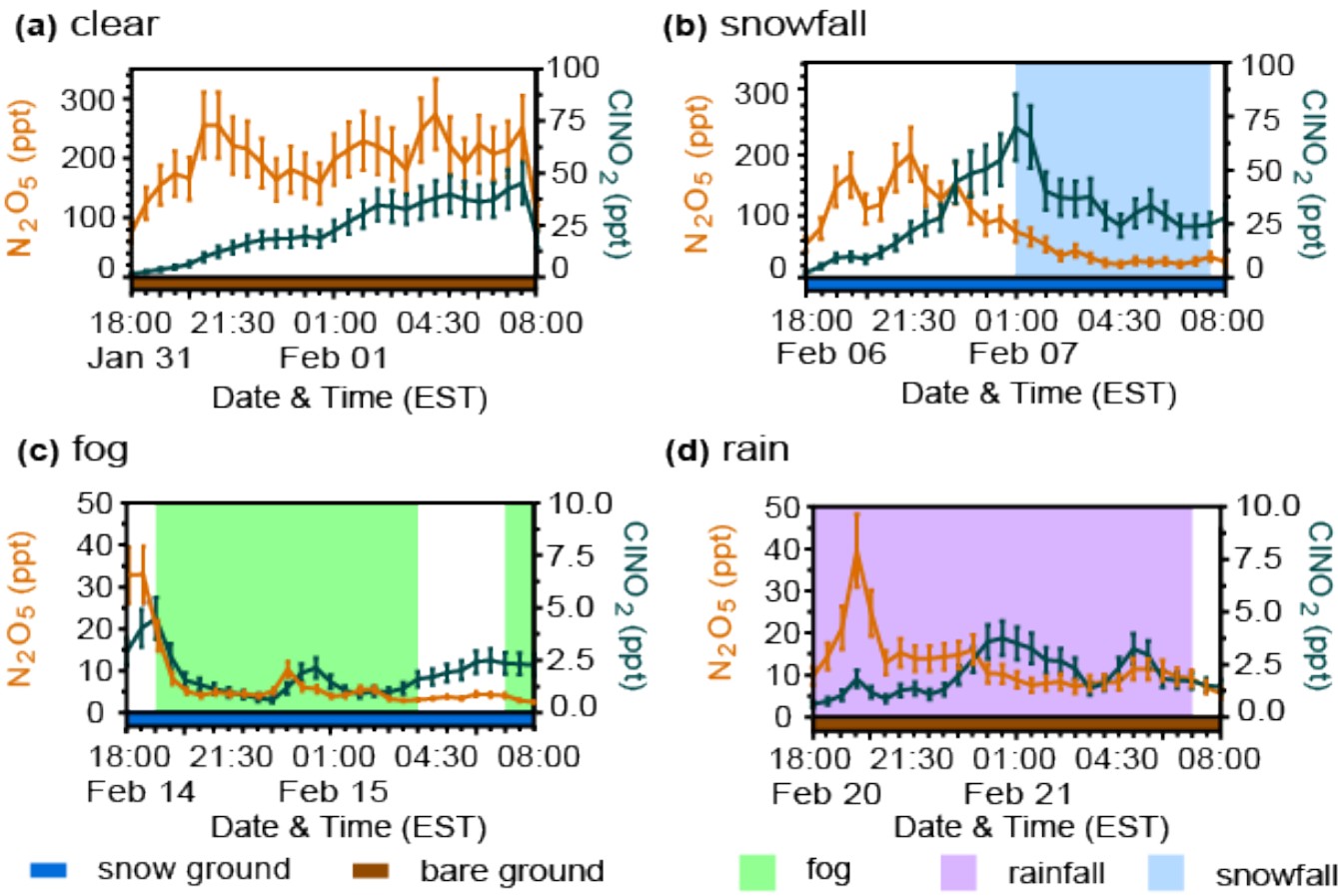

**Figure 3:** Four example case study periods are shown, corresponding to (a) clear conditions, (b) snowfall, (c) fog, and (d) rainfall. The 30-min averaged abundances of $N_2O_5$ *(orange)* and $ClNO_2$ *(dark blue)* are displayed for each case. Error bars represent propagated uncertainties. The shading below the x-axis represents ground cover – snow *(blue)* or bare ground *(brown)*.

To discuss changes in gas-phase concentrations during precipitation and fog, we apply the concept of solution equilibrium to the surface layer of a drop (i.e. a rain or fog droplet) in terms of a local equilibrium between the analyte in the gas-phase and the analyte dissolved in the surface layer (Pruppacher and Klett, 1997). This equilibrium can then be described using Henry's law and Henry's law constants ($K_H$). For $N_2O_5$, fast, irreversible hydrolysis is assumed, equivalent to an infinite effective $K_H$ (Jacob, 1986; Sander, 2015). For $ClNO_2$, the $K_H = 4.5 \times 10^{-4}$ mol m$^{-3}$ Pa$^{-1}$ at standard temperature (Frenzel et al., 1998; Sander, 2015), showing little variation between ~278 and 294 K. Converting the $K_H$ for $ClNO_2$ to its dimensionless Henry solubility (also called the air-water partitioning coefficient, $K_{AW}$), as

in Sander (2015), gives a unitless ratio between the aqueous and gas phases of >1 at temperatures above freezing. This means, at equilibrium, $ClNO_2$ is expected to be more abundant in the aqueous-phase than in the gas-phase. The fast irreversible hydrolysis assumed for $N_2O_5$ makes it more water soluble than $ClNO_2$; therefore, scavenging by liquid droplets is expected for both gas-phase $N_2O_5$ and $ClNO_2$, but to a greater extent for $N_2O_5$. Here, we examine the fog and rainfall case studies to characterize the effects of scavenging by aqueous droplets on $N_2O_5$ and $ClNO_2$ abundance. Variations in $N_2O_5$ and $ClNO_2$ over the course of the nights are likely due to variability in fog/rainfall that were not resolved by the time resolution of the reported weather conditions, which also do not reflect precipitation rates or fog concentrations that would be expected to vary through the nights.For the fog case night of Feb 14-15, fog was present from 19:00-04:00 and 07:00-08:00 (**Fig. 3c**). This case had an average u* of $0.18\pm0.02$ m s$^{-1}$ (campaign average u* was $0.162\pm0.007$ m s$^{-1}$ during nighttime fog) and snow-covered ground. $N_2O_5$ mole ratios decreased rapidly from the maximum of 32 ppt at 18:00 and fell to a local minimum of 2.3 ppt at 22:30; it then remained low in abundance (<10 ppt) for the rest of the night, reaching its true minimum of 1.1 ppt at 03:30. $ClNO_2$ mole ratios reached the maximum of 4.5 ppt at 19:00 and then decreased coincident with the appearance of fog and remained low in abundance (<3 ppt) for the rest of the night, reaching its minimum of 0.6 ppt at 23:00. Considering the first hour after the fog onset (19:00-20:00), $N_2O_5$ mole ratios decreased from 16.6 ppt to 3.4 ppt (decrease of 13.2 ppt or 80%) and $ClNO_2$ mole ratios decreased from 4.5 ppt to 1.6 ppt (decrease of 2.9 ppt or 64%).

Similarly, the rainfall case night of Feb 20-21 was characterized by rainfall from 18:00-07:00 and bare ground (**Fig. 3d**). While sonic anemometer data were unavailable on this night, elevated wind speeds of 2.2-8.9 m s$^{-1}$ (average=$5.0\pm0.5$ m s$^{-1}$) (**Fig. S4** and **Table S2**) are consistent with increased turbulence, with u* likely greater than 0.25 m s$^{-1}$ for the duration of the night (**Fig. S5**). $N_2O_5$ mole ratios decreased rapidly from the maximum of 40 ppt at 19:30, stabilized at ~15 ppt from 20:30-00:00, and then decreased again to ~10 ppt until 08:00. $ClNO_2$ mole ratios reached the maximum of 3.7 ppt at 00:30, with a second local maximum of 3.0 ppt at 05:30; $ClNO_2$ abundance was <2 ppt before 23:30 and after 06:00. The observations during the fog and rainfall case studies reinforce the trends observed for the campaign averages (**Fig. 2-3**) and illustrate the importance of scavenging by liquid droplets.

The snowfall case night of Feb 06-07 was characterized by snowfall from 01:00-07:30 (**Fig. 3b**), an average u* of 0.06±0.01 m s$^{-1}$ (campaign average u* was 0.129±0.004 m s$^{-1}$ during nighttime snowfall), and snow-covered ground. $N_2O_5$ mole ratios reached the maximum of 201 ppt at 21:30 and then gradually decreased throughout the rest of the night; it reached its minimum of 22 ppt at 04:00 and then remained low in abundance (22-34 ppt). $ClNO_2$ mole ratios reached the maximum of 70 ppt at 01:00, the same time that snowfall began, and then decreased steadily to the minimum of 24 ppt at 04:30, after which it also remained low in abundance (24-34 ppt). Considering the first hour after snowfall onset (01:00-02:00), $N_2O_5$ mole ratios decreased from 74.8 ppt to 53.6 ppt (decrease of 21.2 ppt or 28%) and $ClNO_2$ mole ratios decreased from 69.8 ppt to 40.3 ppt (decrease of 29.5 ppt or 42%).

The observations during the snowfall case are also consistent with campaign-wide observations (**Fig. 2-3**). In comparison to the clear case, the snowfall case night shows that $N_2O_5$ mole ratios were generally lower during snowfall (by 2.1 times, on average), whereas $ClNO_2$ mole ratios were typically higher during snowfall (by 1.4 times, on average). Even though the clear case study had the highest mole ratios of $N_2O_5$, the snowfall case study had the highest mole ratios of $ClNO_2$ (**Fig. 3** and **Table S2**). The clear and snowfall case studies differed in both ground cover and air turbulence, with lower friction velocity (average=0.06±0.01 m s$^{-1}$) and snow-covered ground observed during the snowfall case and intermediate friction velocity (average=0.16±0.01 m s$^{-1}$) and bare ground observed during the clear case study. Additional effects on the abundances of $N_2O_5$ and $ClNO_2$ are further investigated in the following sections.

### 3.3 Effects of turbulence

Turbulent mixing (quantified here using friction velocity, u*, **E1**) affects abundances of surface-level trace gases (Stull, 1988). Stronger turbulent mixing promotes vertical transport, and weaker turbulent mixing keeps trace gases near the ground (Stull, 1988). Turbulence regimes were divided within the context of our study to allow subsequent analysis by binning with sufficient data in each bin. Here, lower turbulence refers to u* < 0.1 m s$^{-1}$, higher turbulence is u* > 0.25 m s$^{-1}$, and mid-turbulence refers to 0.1 < u* < 0.25 m s$^{-1}$. Lower turbulence occurred 39% of the time, mid-turbulence occurred 42% of the time, and higher turbulence occurred 14% of the time (**Fig. S3**). For context, typical u* values range

from near 0 m s$^{-1}$ during calm conditions to 1 m s$^{-1}$ during strong winds; moderate wind values often have u* values near 0.5 m s$^{-1}$ (Stull, 2017). Lower friction velocity, in general, was observed during our study, which focuses on nighttime measurements during winter. We investigate the effects of atmospheric turbulence on the abundances of ClNO$_2$ and N$_2$O$_5$ by comparing lower (u* < 0.1 m s$^{-1}$) and higher turbulence (u* > 0.25 m s$^{-1}$) periods across 30 min averaged periods for a full diel cycle during the entire campaign (**Fig. 4**). Periods of snowfall, fog, and rainfall were included in this analysis due to the relationships which exist between weather events and friction velocity; for example, snowfall occurred most often during lower turbulence conditions, while rainfall occurred most often during higher turbulence conditions, the effects of which are further discussed in *Sect. 3.5*.

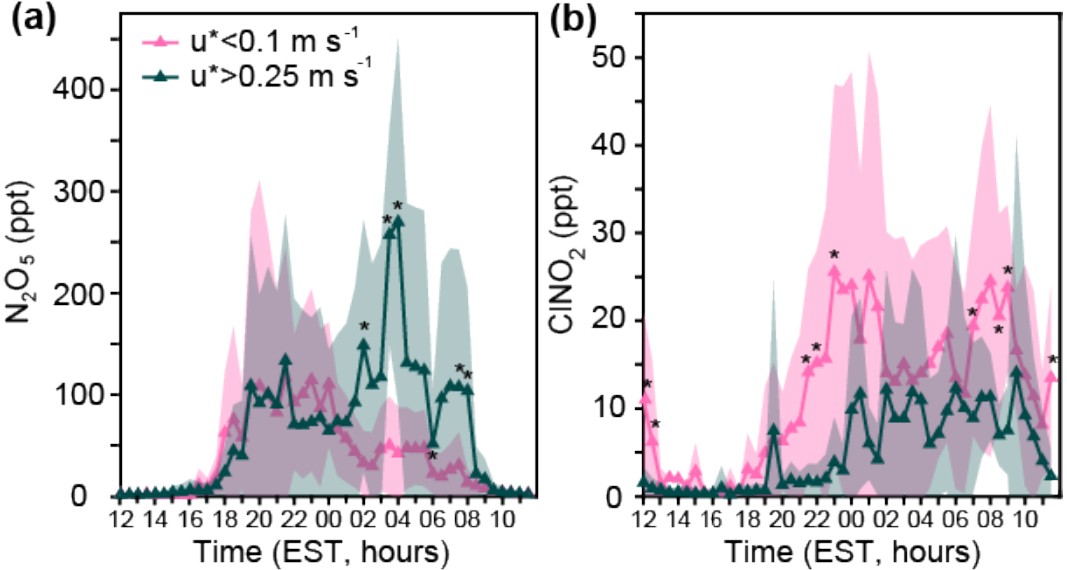

**Figure 4.** Campaign-wide diel patterns of 30 min averaged (a) N$_2$O$_5$ and (b) ClNO$_2$, binned by lower (u*<0.1 m s$^{-1}$) and higher (u*>0.25 m s$^{-1}$) friction velocities. Shading represents one standard deviation. Asterisks represent statistically significant (t-test) differences at the p<0.05 level between the lower and higher friction velocity bins for each 30 min period from January 20-February 24 (excluding February 20-21 when the sonic anemometer was not operational). The number of 30 min time periods, from 18:00-08:00, is reported as n. Lower turbulence occurred 39% of the time (n=391) and higher turbulence occurred 14% (n=137) of the time; sonic anemometer data were unavailable for 5% (n=53) of nighttime periods.

Significantly higher (p<0.05, t-test) N$_2$O$_5$ mole ratios were observed under higher turbulence conditions at 02:00, 03:30, 04:00, 06:00, and 07:30, and 08:00 (**Fig. 4a**). These statistically significant

time points correspond to, on average, 5.9 times higher $N_2O_5$ mole ratios during higher turbulence conditions, in comparison to lower turbulence conditions. Considering the entire period of 02:00-08:00, $N_2O_5$ mole ratios were 4.0 times higher, on average, during higher turbulence conditions in comparison to lower turbulence conditions. Before 02:00, no statistically significant differences were observed in $N_2O_5$ abundance between higher and lower turbulence conditions, suggesting that titration of $NO_3$ ($N_2O_5$ precursor, R3) by NO was not significant during these time periods. Considering the nighttime period as a whole (18:00-08:00), $N_2O_5$ mole ratios were higher by 24±4ppt (1.6-fold) during higher turbulence, in comparison to lower turbulence conditions. For context in relation to the various weather conditions, higher turbulence ($u^*>0.25$ m s$^{-1}$) was present for 13%, 9%, 19%, and 17% of the time during clear, snowfall, fog, and rainfall conditions, respectively (**Fig. S3**).

$NO_3$, a reactant necessary to produce $N_2O_5$ (**R3**), is sensitive to changes in NO and $O_3$ levels; in particular, titration of $NO_3$ by NO (**R7**) is an important loss process at night and results in lower $N_2O_5$ production (Asaf et al., 2010). Therefore, when NO is emitted and confined near the ground in the stable nocturnal boundary layer, $NO_3$ has a short near-surface lifetime, thereby limiting $N_2O_5$ levels (Brown et al., 2007; Wang et al., 2006). Such stable conditions are associated with nocturnal temperature inversions, which can be observed during wintertime in the mid-latitudes (Leblanc and Hauchecorne, 1997). As expressed by kinematic heat flux less than 0 K m s$^{-1}$, a nocturnal temperature inversion was observed every night of the study (**Fig. S7**). As expected during more stable conditions, reduced $N_2O_5$ mole ratios were observed during nighttime lower turbulence ($u^*<0.1$ m s$^{-1}$) compared to higher turbulence ($u^*>0.25$ m s$^{-1}$) periods (average $N_2O_5$ mole ratios of 40±2 ppt and 64±3 ppt, respectively).

Vehicle $NO_x$ emissions from the nearby roadway location ~80 m away (McNamara et al., 2021) are suggested to control the magnitude of the nighttime titration effect at the field site, as few time periods overnight were statistically different in $O_3$ mole ratios between the lower and higher turbulence conditions, on average (**Fig. S8**). However, despite 39% of the nighttime periods being characterized by lower turbulence ($u^*<0.1$ m s$^{-1}$) (**Fig. S3**), $N_2O_5$ mole ratios during the full campaign ranged from 0.15-702 ppt (mean 44±4 ppt) during nighttime, resulting in the observed $ClNO_2$ production even under lower turbulence conditions. We explored the loss of $NO_3$ to reaction with VOCs (e.g. **R6**) using a box numerical model (*Sect. S1*, **Fig. S14**). These modeling results show that during the four case nights, which

had varying temperatures, friction velocities, and ground cover (**Fig. 3, Table S2**), $NO_3$ is simulated to be lost primarily through formation of $ClNO_2$ and $HNO_3$ (**Fig. S14**), rather than reaction with VOCs.

In contrast to its precursor $N_2O_5$, $ClNO_2$ shows significantly higher ($p<0.05$, t-test) average mole ratios under lower turbulence ($u^*<0.1$ m s$^{-1}$) conditions at 21:30, 22:00, 23:00, and 07:00 (**Fig. 4b**). These statistically significant time periods correspond to an average 6.3 times higher $ClNO_2$ mole ratio during lower turbulence conditions, in comparison to higher turbulence conditions. Considering the entire period of 21:30-07:30, $ClNO_2$ mole ratios were 3.6 times higher, on average, during lower turbulence conditions in comparison to higher turbulence conditions. Considering the nighttime period as a whole (18:00-08:00), $ClNO_2$ mole ratios were higher by 7±1 ppt (2.6-fold) during lower turbulence, in comparison to higher turbulence conditions.

In summary, average $N_2O_5$ mole ratios were significantly higher ($p<0.05$) at six different 30 min time periods, corresponding to 5.9 times higher $N_2O_5$ mole ratios during higher turbulence conditions, in comparison to lower turbulence conditions. The reduced $N_2O_5$ mole ratios observed under lower turbulence conditions are likely due to the short lifetime of $NO_3$ ($N_2O_5$ precursor, R3) when vehicle $NO_x$ is emitted into the stable boundary layer, as observed in previous studies (Brown et al., 2007; Wang et al., 2006). However, average $ClNO_2$ mole ratios were significantly higher ($p<0.05$) during four different 30 min time periods, corresponding to 6.3 times higher $ClNO_2$ mole ratios during lower turbulence conditions, in comparison to higher turbulence conditions. This points to a likely surface source of $ClNO_2$ upon surface deposition of $N_2O_5$. Therefore, in Section 3.4, we investigated the influence of ground cover (Section 3.4).

### 3.4 Effects of ground cover

There were no statistically significant ($p<0.05$, t-test) differences in the average abundances of $N_2O_5$ over the diel period for snow-covered vs bare ground (**Fig. 5a**). This is consistent with measurements of similar net negative (deposition) fluxes of $N_2O_5$ over both snow-covered and bare ground (McNamara et al., 2021). The nighttime average mole ratios of $N_2O_5$ were 70±5 ppt and 68±4 ppt over snow-covered and bare ground, respectively. In contrast, **Figure 5b** shows significantly higher ($p<0.05$, t-test) average $ClNO_2$ mole ratios observed over snow-covered ground at 19:30-22:00, 23:00-

00:00, 01:00-01:30, 03:00, and 07:30. These statistically significant time points correspond to, on average, 3.5 times higher $ClNO_2$ mole ratios over snow-covered ground, in comparison to bare ground. Considering the entire period of 19:30-07:30, $ClNO_2$ mole ratios were 2.8 times higher, on average, over snow covered ground in comparison to bare ground. This is consistent with measurements of typical net positive (production) fluxes of $ClNO_2$ over snow-covered ground, and with field-based chamber experiments showing that $ClNO_2$ can be produced from the reaction of $N_2O_5$ on the saline snowpack (McNamara et al., 2021). The nighttime average mole ratios of $ClNO_2$ were 14.9±0.8 ppt and 7.0±0.5 ppt over snow-covered and bare ground, respectively.

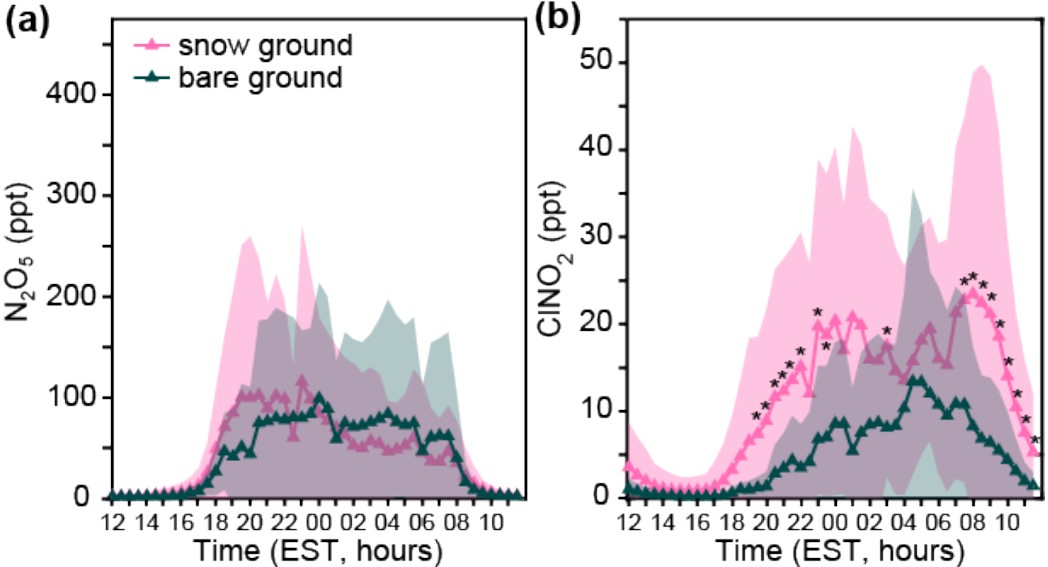

**Figure 5.** Diel patterns of 30 min averaged mole ratios of (a) $N_2O_5$ and (b) $ClNO_2$ binned by snow-covered and bare ground conditions from January 20 to February 24. Shading represents one standard deviation. Asterisks represent statistically significant (t-test) differences at the $p < 0.05$ level between snow-covered and bare ground for each 30 min time period. The ground was snow-covered 57% [20 d] of the study and was bare for 43% [15 d] of the study.

To summarize, there were no statistically significant ($p<0.05$) differences in average $N_2O_5$ mole ratios over the diel period for snow-covered versus bare ground. Yet, significantly higher ($p<0.05$) average $ClNO_2$ mole ratios were observed over snow-covered ground for 11 (of 28) nighttime 30 min time periods, corresponding to 3.5 times higher $ClNO_2$ mole ratios over snow-covered ground, in comparison to bare ground. During the same field campaign, net positive (production) fluxes of $ClNO_2$ were measured over snow-covered ground, and field-based chamber experiments showed that $ClNO_2$ can

be produced from the reaction of $N_2O_5$ on the saline snowpack (McNamara et al., 2021). The observed

enhancement of $ClNO_2$ over snow-covered ground results herein suggests that snowpack $ClNO_2$ production was a frequent and significant occurrence across the field campaign (e.g. enough to influence the campaign-wide average results). We investigate the effects of other parameters (e.g. $PM_{2.5}$ $Cl^-$, $NO_3^-$, temperature, relative humidity, $O_3$ concentration, aerosol surface area, and pressure) in Section 3.5.

## 3.5 Competing effects of environmental conditions

Many of the environmental conditions discussed (precipitation/fog, turbulence regimes, and snow-covered/bare ground) occur simultaneously, and as a result, are difficult to discuss in isolation. Higher mole ratios of $N_2O_5$ were observed under higher turbulence conditions ($u^* > 0.25$ m s$^{-1}$) (Section 3.3), which occurred most frequently (67%) over bare ground (**Fig. S3**). In contrast, higher mole ratios of $ClNO_2$ were observed under lower turbulence conditions ($u^* < 0.25$ m s$^{-1}$) (Section 3.3), which occurred

most frequently (73%) over snow-covered ground (**Fig. S3**). For select nights when vertical profile experiments were conducted by McNamara et al. (2021) during the same campaign, no statistically significant difference ($p = 0.48$) was observed for $N_2O_5$ deposition fluxes over bare ground versus snow-covered ground. Lower turbulence ($u^* < 0.1$ m s$^{-1}$) and snow-covered ground were observed simultaneously for 24%, 48%, 26%, and 2% of the time during clear, snowfall, fog, and rainfall

conditions, respectively (**Fig. S3**). The prevalence of lower turbulence and snow-covered ground during snowfall likely also contributes to the result that mole ratios of $ClNO_2$ were highest on average during snowfall (**Fig. 2**). These trends are consistent with snowpack $ClNO_2$ production, as also evidenced by the positive (upward) $ClNO_2$ fluxes observed over snow-covered ground and negative (downward) $ClNO_2$ fluxes observed over bare ground, during separate vertical profile experiments during the same campaign

McNamara et al. (2021).

Given that multiple environmental factors that control $N_2O_5$ and $ClNO_2$ mole ratios are changing across the various weather conditions, we used a box numerical model, described in Section S1, to explore the variations in $N_2O_5$ and $ClNO_2$ abundances that can be attributed to changes in temperature, pressure, $O_3$ mole ratios, and aerosol surface area across the four case study nights. This numerical model does not

consider the impacts of fog, rainfall, snowfall, ground cover, turbulence, or advection. Note that no

relationship was observed between wind direction or wind speed and mole ratios of $N_2O_5$ or $ClNO_2$ (**Fig. S13**), suggesting limited advection influence. Further, under the low wind speed conditions of the campaign (nighttime median=1.0 m s$^{-1}$), the gas mole ratios are expected to be higher in response to decreased atmospheric dispersion of gases and emissions/deposition from nearby sources/sinks. Therefore, we mainly attribute differences between calculated and measured $N_2O_5$ and $ClNO_2$ abundances primarily to the effects of non-parameterized meteorological processes (e.g., wet deposition and fog droplet scavenging). The model results (**Fig. 6, S14-16**) are discussed in detail in Section S2. Importantly, we conclude that variations in temperature, pressure, $O_3$ mole ratios, and aerosol surface area between the different case studies are insufficient to explain the significant differences in $N_2O_5$ and $ClNO_2$ mole ratios observed between these case study nights and point to the importance of other processes, including scavenging, discussed in this manuscript.

For the clear snowfall, fog, and rainfall case study nights, simulated $N_2O_5$ mole ratios averaged 150 ppt, 190 ppt, 140 ppt, and 380 ppt, respectively, during the last 4 h of the simulation (hours 10-14, 04:00-08:00 EST, to account for model spin-up and stabilization) (**Fig. 6**). In comparison, the maximum observed $N_2O_5$ mole ratios were 274 ppt, 34.2 ppt, 2.7 ppt, and 11.4 ppt from 04:00-08:00 EST during the clear, snowfall, fog, and rainfall cases respectively (**Fig. 6**). While plausible scenarios of [$NO_2$] and $N_2O_5$ uptake could simulate the observed $N_2O_5$ mole ratios (**Fig. S15**), the model scenario corresponding to previous work in wintertime Ann Arbor, MI (McNamara et al., 2020) underpredicts the average $N_2O_5$ mole ratio (229 ppt) by 42% . However, in contrast, the model drastically over-predicted $N_2O_5$ mole ratios for both the fog case (by ~50 times) and rainfall case (by ~30 times), and also over-predicted $N_2O_5$ mole ratios during the snowfall case (by ~5 times). As discussed in Section S2, realistic model conditions could not simulate the observed $N_2O_5$ mole ratios for the fog, rainfall, and snowfall cases. This supports scavenging as a missing $N_2O_5$ sink, with this being most significant during fog and rainfall, and potentially also contributing during snowfall.

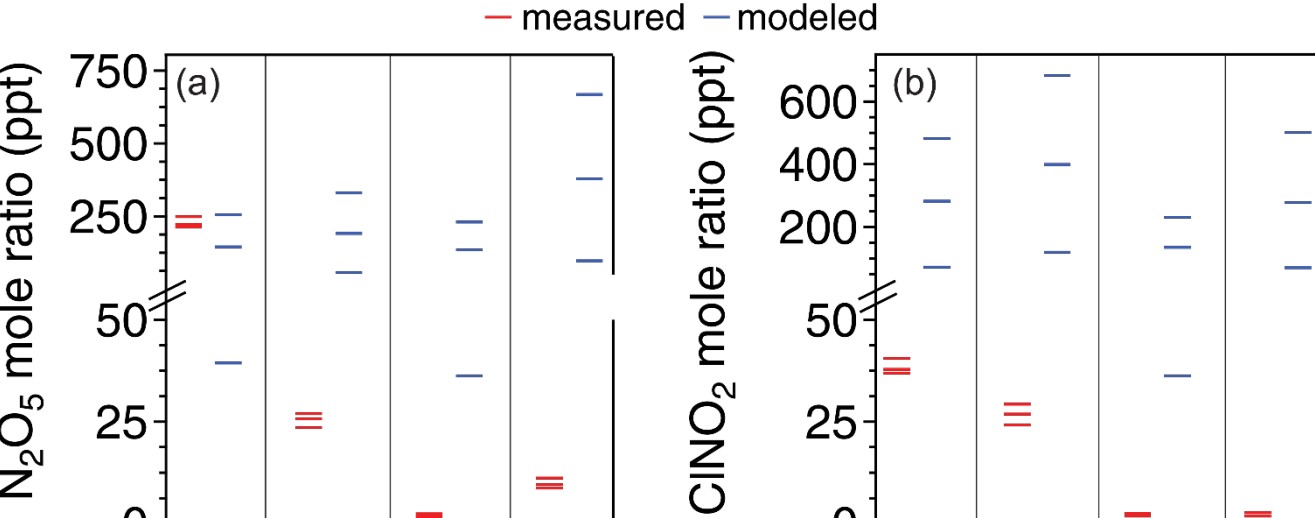

**Figure 6:** A comparison of measured (red) vs modeled (blue) mole ratios of $N_2O_5$ (a) and $ClNO_2$ (b) for each case study night. For measured values, the minimum, average, and maximum mole ratios from 04:00 – 08:00 EST of each case study (Fig. 3) are shown. For modeled values, the average mole ratios from the last 4 h of each 14 h simulation are shown; model inputs for the case studies are described in *Sect. S1*. Here, we hold the $\gamma*\varphi$ product ($N_2O_5$ uptake*$ClNO_2$ yield) constant at 0.0037, and show the model outputs when $[NO_2]$ = 9.4 ppb, 31 ppb, and 59 ppb, respectively; $[NO_2]$ = 9.4 ppb produced the lowest modeled values of $N_2O_5$ and $ClNO_2$, and $[NO_2]$ = 59 ppb produced the highest modeled values of $N_2O_5$ and $ClNO_2$. For context, McNamara et al. (2020) previously reported a modeled median $[NO_2]$ of 31 ppb and $\gamma*\varphi$ product constant of 0.0037 for wintertime Ann Arbor, MI.

Considering the entire SNACK field campaign, temperature was statistically significantly different between clear conditions and snowfall, fog, and rainfall, respectively ($p<0.05$, t-test) (**Fig. 7c**). The average nighttime temperatures were 265.8±0.2 K, 270.8±0.3 K, 276.7±0.2 K, and 282.1±0.2 K during snowfall, clear conditions, fog, and rainfall, respectively. Since lower temperatures favor $N_2O_5$ production in its thermal equilibrium (**R3**) (Asaf et al., 2010; Wagner et al., 2013), and because snowfall conditions had the lowest average temperature (**Fig. 7c**), we would expect $N_2O_5$ to be highest in abundance during snowfall if other processes did not dominate. In contrast, the measurements showed the highest average $N_2O_5$ mole ratios during clear conditions (**Fig. 2**), highlighting the importance of other effects, including wet scavenging. Further, as shown in the case study model simulations, discussed above, that did not consider scavenging, simulated $N_2O_5$ mole ratios were highest during the rainfall case

due to lower aerosol surface area concentrations, and second highest during the snowfall case because of the temperature effect (**Fig. 7**). Therefore, we conclude that temperature alone cannot explain the significant differences in $N_2O_5$ mole ratios between the clear, fog, rainfall, and snowfall conditions.

Relative humidity was also statistically significantly different between clear conditions and snowfall, fog, and rainfall, respectively ($p<0.05$) across the SNACK field campaign (**Fig. 7d**). The average nighttime RH values were 75.0±0.5%, 83.0±0.3%, 90.2±0.4%, and 93.7±0.3% during clear conditions, snowfall, rainfall, and fog, respectively. Higher RH typically increases $N_2O_5$ partitioning from the gas to aqueous phases (e.g. Osthoff et al., 2006; Sommariva et al., 2009; Wood et al., 2005). Indeed,

the pattern of $N_2O_5$ abundance was anticorrelated with RH (**Fig. 2** and **Fig. 7d**). This reinforces that $N_2O_5$ heterogeneous uptake is strongly RH dependent (Bertram et al., 2009; Davis et al., 2008; Evans and Jacob, 2005; Griffiths and Cox, 2009; Hallquist et al., 2003), with enhanced uptake and removal occurring when RH and aerosol liquid water content are high.

    The box model overestimated $ClNO_2$ mole ratios for the clear case (by ~6 times), despite lower

simulated $N_2O_5$ mole ratios compared to modeled values (by ~42%) (**Fig. 6**), as discussed in the Section S2. Since the chosen values for [$NO_2$], $N_2O_5$ uptake, and $ClNO_2$ yield corresponded to previous work in wintertime Ann Arbor, MI (McNamara et al., 2020), this points to the variability and need to better constrain $N_2O_5$ uptake and $ClNO_2$ yield, as highlighted previously by McDuffie et al. (2018). However, realistic model conditions could be chosen to simulate the observed clear case $N_2O_5$ and $ClNO_2$ mole

ratios (**Fig. S15 and S16**). In contrast to the clear case, realistic [$NO_2$], $N_2O_5$ uptake, and $ClNO_2$ yield values could not be chosen to simulate the observed $ClNO_2$ mole ratios, similar to the result for $N_2O_5$ mole ratios, discussed above. Average simulated $ClNO_2$ mole ratios were 400 ppt, 140 ppt, and 280 ppt for the snowfall, fog, and rainfall cases, respectively, during the last 4 h of the simulation (**Fig. 6**). In comparison, the maximum observed $ClNO_2$ mole ratios were 33.6 ppt, 2.5 ppt, and 3.2 ppt from 04:00-

08:00 EST of the snowfall, fog, and rainfall cases, respectively (**Fig. 6**). Further, the model drastically overpredicted $ClNO_2$ mole ratios during the fog case (by ~50 times, similar to the $N_2O_5$ mole ratio overprediction) and rainfall case (by ~90 times, as compared to ~30 times over-prediction of $N_2O_5$ mole ratios), but also over-predicted $ClNO_2$ mole ratios during the snowfall case (by ~12 times, as compared to ~5 times for $N_2O_5$ mole ratios). The similar overprediction of $N_2O_5$ and $ClNO_2$ during fog supports fog

droplet scavenging of $N_2O_5$, in particular, as a missing sink in the model. The higher overprediction of $ClNO_2$ mole ratios, compared to $N_2O_5$, during the rainfall case, in particular, suggests that $ClNO_2$, in addition to $N_2O_5$, likely undergoes scavenging/wet deposition.

We also investigated $N_2O_5$ and $ClNO_2$ levels in the context of observed $PM_{2.5}$ $Cl^-$ and $NO_3^-$ concentrations. The averages for these parameters are given for clear conditions and each type of weather
event in **Table 1**, with additional data provided in **Table S2**. As shown by Bertram and Thornton (2009), both $N_2O_5$ uptake and the product yield of $ClNO_2$ are expected to increase with increasing particulate chloride concentrations. The effects of increased particulate chloride are two-fold, with less $N_2O_5$ expected to remain in the gas-phase due to the increased uptake, and a higher $ClNO_2$ abundance expected because of the higher product yield. $PM_{2.5}$ $Cl^-$ concentrations were not statistically significantly different
between snowfall and clear conditions ($p=0.96$, t-test), between snowfall and rainfall ($p=0.11$), or between clear and rainfall conditions ($p=0.10$) (**Fig. 7a**).

$PM_{2.5}$ $Cl^-$ concentrations were statistically significantly higher during fog, in comparison to clear conditions ($p<0.05$), with the average concentration during fog higher by $0.20\pm0.01$ $\mu g$ $m^{-3}$ (1.8 times) on average. Although total submicron aerosol number concentrations were not statistically significantly
different between clear and fog conditions ($p=0.88$), submicron aerosol surface area concentrations were significantly higher ($p<0.05$) during fog compared to clear conditions, by $52\pm7$ $\mu m^2$ $cm^{-3}$ (1.3 times) with respect to campaign averages (**Fig. S11-S12**). $N_2O_5$ uptake is expected to increase with increasing aerosol surface area concentration (Bertram and Thornton, 2009), but despite elevated $PM_{2.5}$ $Cl^-$ and aerosol surface area concentrations during fog, average $ClNO_2$ abundance was lower during fog in comparison
to clear conditions (**Fig. 2**). We expect that, during fog, elevated RH (**Fig. 7d**) has a greater impact on $ClNO_2$ abundance than $PM_{2.5}$ $Cl^-$ concentration or aerosol surface area concentration. Production of particle-phase chloride, presumed to be from uptake of gas-phase HCl, has been observed previously during fog/haze events in highly polluted urban India (Gunthe et al., 2021) and near an incinerator (Johnson et al., 1987). However, for this study in Kalamazoo, MI, road salting seems more plausible as
the dominant source of increased $PM_{2.5}$ $Cl^-$ during wintertime fog.

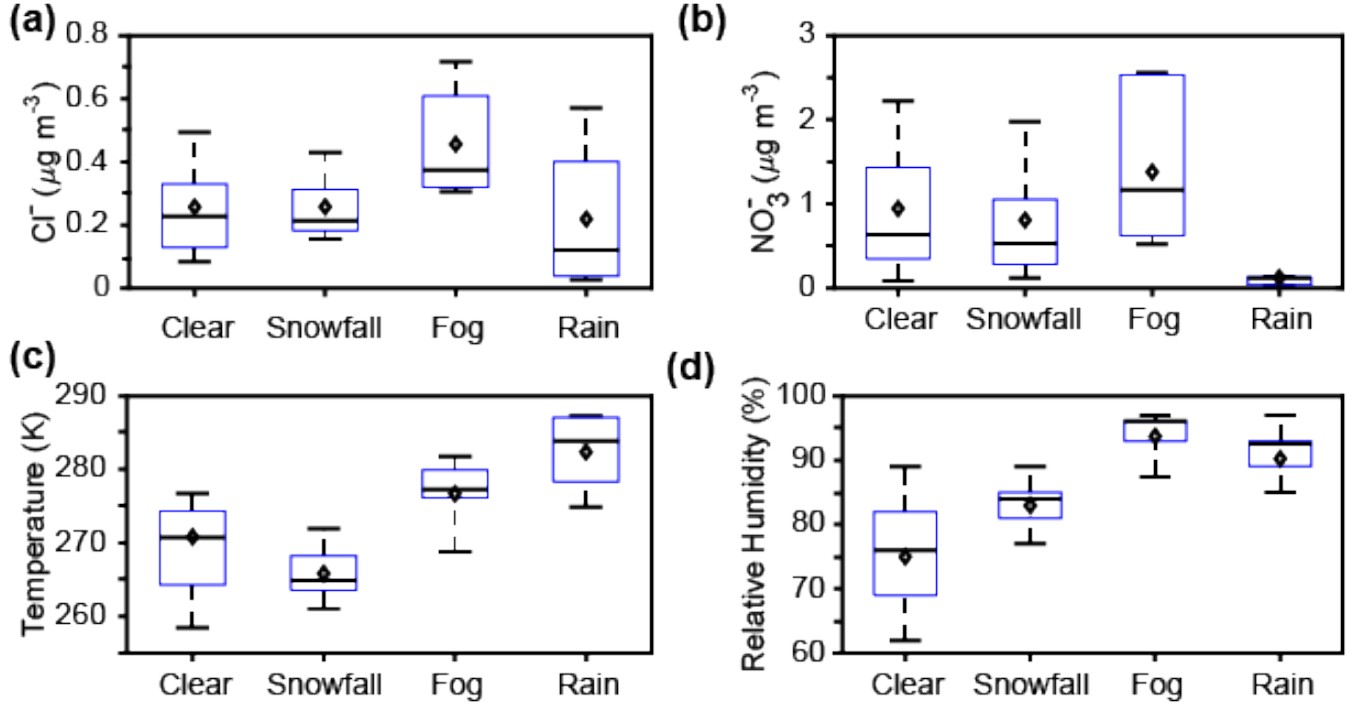

**Figure 7.** Box plots showing 30 min averaged PM$_{2.5}$ (a) chloride and (b) nitrate concentrations, (c) air temperatures and (d) relative humidity values during clear conditions and weather events (snowfall, fog, and rain). Bars represent the 10$^{th}$, 50$^{th}$, and 90$^{th}$ percentiles, boxes represent the 25$^{th}$ and 75$^{th}$ percentiles, and diamonds represent the means. Only nighttime data, between 18:00 and 08:00 EST, are included.

N$_2$O$_5$ uptake results in particulate nitrate production; however, the efficiency of N$_2$O$_5$ uptake to particles decreases with increasing particulate nitrate concentrations (Bertram and Thornton, 2009). PM$_{2.5}$ NO$_3^-$ concentrations were not statistically significantly different between snowfall and clear conditions (p=0.08). PM$_{2.5}$ NO$_3^-$ concentrations during rain were statistically significantly lower, in comparison to clear conditions (p<0.05), with average concentrations lower by 0.82±0.04 µg m$^{-3}$ (7.5 times) (**Fig 6b**). This is attributed to increased scavenging and wet deposition of nitrate during rainfall, compared to snowfall, which is consistent with previous observations and calculations of scavenging coefficients for nitrate during rainfall and snowfall in winter in New York (Sperber and Hameed, 1986).

Particles rich in nitrate have been observed previously in the droplet mode (0.8–0.9 µm) during fog events; these particles form following fog droplet evaporation after nitrate production from HNO$_3$ and N$_2$O$_5$ uptake (Dall'Osto et al., 2009; Ge et al., 2012). In contrast to rain and snowfall, PM$_{2.5}$ NO$_3^-$

concentrations were statistically significantly higher during fog, in comparison to clear conditions (p<0.05), by 0.43±0.06 µg m$^{-3}$ (160±20 ppt; 1.5 times) (**Figs. 6b** and **S10**). The increase in PM$_{2.5}$ NO$_3^-$ is likely, in part, the result of heterogeneous uptake and hydrolysis of N$_2$O$_5$ (Brown et al., 2004; Osthoff et al., 2006), consistent with our observation of the lower average N$_2$O$_5$ mole ratios during fog (**Fig. 2**). On average, N$_2$O$_5$ was 76±5 ppt lower during fog compared to clear conditions (**Figs. 2** and **S10**). While this difference is not completely attributable to N$_2$O$_5$ uptake, it would correspond to a nitrate concentration of 0.21 µg m$^{-3}$. In addition to N$_2$O$_5$, gas-phase HNO$_3$ uptake likely also contributed to the increased PM$_{2.5}$ NO$_3^-$ observed during fog. Due to its high solubility, HNO$_3$ is predicted to be efficiently scavenged by fog droplets (>90-100% removal) (Ervens, 2015). However, due to the limited HNO$_3$ data available (**Fig. S9**), a quantitative evaluation of HNO$_3$ contribution to nitrate production was not possible. It is likely that both N$_2$O$_5$ and HNO$_3$ uptake, followed by aqueous-phase nitrate formation, led to the increased PM$_{2.5}$ NO$_3^-$ observed during fog.

## 4 Conclusions

We examined the impacts of precipitation (rain, snowfall) and fog, atmospheric turbulence, and ground cover (snow-covered vs bare) on near-surface (~1.5 m above ground) N$_2$O$_5$ and ClNO$_2$ observed during January to February 2018 in Kalamazoo, Michigan. While N$_2$O$_5$ was observed during all nights of the campaign, N$_2$O$_5$ mole ratios were lowest during periods of lower turbulence (u*<0.1 m s$^{-1}$) due to titration of NO$_3$ and O$_3$ by NO in the stable nocturnal boundary layer. N$_2$O$_5$ mole ratios were not statistically significantly different over bare versus snow-covered ground. ClNO$_2$ mole ratios were highest during periods of lower turbulence and snow-covered ground. This is consistent with N$_2$O$_5$ depositing and reacting with the chloride-containing snowpack to produce ClNO$_2$. Indeed, vertical gradient measurements during the same study showed N$_2$O$_5$ deposition and an average positive (production) ClNO$_2$ flux over snow-covered ground, and snow chamber experiments showed that synthesized N$_2$O$_5$ reacted with the local saline snow to produce ClNO$_2$ (McNamara et al., 2021). This finding is also consistent with the laboratory study by Lopez-Hilfiker et al. (2012), which showed that N$_2$O$_5$ can react on halide-doped ice surfaces to produce ClNO$_2$. The contribution of the snowpack as a common ClNO$_2$

source across the field campaign has important implications for the vertical distribution of atmospheric chlorine chemistry, which will be examined in a future manuscript through one-dimensional modeling for comparison with chloride-containing aerosol particles that serve as a major $ClNO_2$ source.

On average, both $N_2O_5$ and $ClNO_2$ abundances were lowest during rainfall and fog due to scavenging. While both species are water soluble, $N_2O_5$ undergoes more efficient scavenging by liquid droplets, particularly fog, as expected based on its higher Henry's Law constant and uptake coefficient (Fickert et al., 1998; Gržinić et al., 2017). $N_2O_5$ uptake by fog droplets likely contributed to observed elevated $PM_{2.5}$ $NO_3^-$ during fog events. Little is known about $N_2O_5$ and $ClNO_2$ scavenging by precipitation, supporting the need for further investigation of this process. Overall, our results show that observational and modeling studies of only clear conditions miss important processes including scavenging, fog nitrate production, and the snowpack as a $ClNO_2$ source. This is important as rainfall, fog, and snowfall occurred during 28% of the nighttime periods, representing a significant portion that contributes significantly to the variability observed during this winter study.

## Acknowledgements

This study was supported by the US National Science Foundation Atmospheric Chemistry program (AGS-1738588 and PLR-1417914), an Alfred P. Sloan Foundation Research Fellowship in Chemistry, and the University of Michigan. J.E. acknowledges funding from the Swiss National Science Foundation (155999). We thank Andrew Ault, Nicholas Ellsworth, and Matthew McNamara for assistance in preparing the mobile laboratory, Angela Raso, Peter Peterson, Guy Burke, and Alexa Watson for fieldwork assistance, and Western Michigan University for use of their facilities for this study.

## Data Availability

The CIMS and AIM-IC datasets are archived through PANGAEA: Kulju, Kathryn; Pratt, Kerri A (2021): $N_2O_5$, $ClNO_2$, $PM_{2.5}$ chloride, and $PM_{2.5}$ nitrate in Kalamazoo, Michigan, USA during January-February 2018. PANGAEA, 2021, https://doi.org/10.1594/PANGAEA.933765.

## Author Contributions

KK wrote the manuscript, with feedback from all coauthors. KP designed the study, and SM and JE conducted the measurements and calibrations. KK led data analysis and interpretation, with contributions from SM, JE, QC, JDF, and KP. HK and KK completed the box modeling. JDF assisted with the air turbulence measurements and analysis, in particular. SB coordinated logistics at the field site.

## Competing Interests

The authors declare that they have no conflict of interest.

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
