# Peer review of "Urban inland wintertime $\text{N}_2\text{O}_5$ and $\text{ClNO}_2$ influenced by snow-covered ground, air turbulence, and precipitation"

_Atmospheric Chemistry and Physics, 2021_

## Author Comment (AC1)

**Response to editor and reviewer comments on "Urban inland wintertime N2O5 and ClNO2 influenced by snow-covered ground, air turbulence, and precipitation" (acp-2021-310)**

We kindly thank the reviewers and editor for their helpful feedback. Comments have been addressed, and changes to the manuscript have been made in response to the received comments and for clarification. We believe these revisions have significantly improved the manuscript. Reviewer comments are in black, our responses are in blue, and additions to the manuscript are shown in italics.

**Editor**

I am confident that this manuscript presents a novel set of observations, as the results reported in McNamara et al., 2020, from the same field campaign are focused on manipulation studies, not ambient observations.

However, in general, the reviews identify some useful areas of improvement for the manuscript. In addition, I provide some points the authors may want to consider while revising their manuscript:

I suggest replacing Figure 1 with Figure S2 so that it's clear that the majority of the analyses use the full dataset and not just the case studies used for the scavenging calculations.

Thank you for this suggestion to improve clarity. We switched Figure 1 and Figure S2, which are now Figure S2 and Figure 1, respectively.

Given the importance of scavenging during rainfall, fog (and to a lesser extent snowfall) should those conditions be excluded from the examination of the impacts of turbulence?

Snowfall occurred most often coincident with lower turbulence conditions, and rainfall occurred most often with higher turbulence conditions, as described in Section 3.5 "Competing effects of environmental conditions". This is an important consideration, even though the magnitude of changes in local mole ratios caused by scavenging is larger than the magnitude of changes caused by turbulence effects. To clarify why time periods of rainfall, fog, and snowfall were included in the investigation of turbulence, we added the following text to Section 3.3 "Effects of turbulence": *"Periods of snowfall, fog, and rainfall were included in this analysis due to the relationships which exist between weather events and friction velocity  for example, snowfall occurred most often during lower turbulence conditions, while rainfall occurred most often during higher turbulence conditions, the effects of which are further discussed in Sect. 3.5."*

The titration of O3 and NO3 by NO is invoked as an explanation for the relationship between N2O5 and turbulence – are there NOx measurements to support this hypothesis?

Unfortunately, $NO_x$ measurements are not available from this field campaign. However, we point to other studies (Brown et al., 2007; Wang et al., 2006) which discuss the short lifetime of surface-level $NO_3$ due to fast reactions of ground-level emitted NO, which is discussed in Sect. 3.3.

As one reviewer points out, advection may be an important term in the local budget of these pollutants (indeed this connects to the point above). Can the authors provide evidence that the impacts of locally measured ground cover and turbulence dominate over (possibly correlated) advection, for example based on wind direction analysis? Additional evidence could be provided from a back-of-the-envelope calculation of the area over which the locally measured vertical exchange/surface chemistry processes would need to occur in order to meaningfully influence the budgets of N2O5 and ClNO2.

Wind speed and direction analysis has been added, as shown in Figure S13, included below. The caption reads: "Polar plots showing 30 min averaged wind direction (angle, degrees), 30 min averaged wind speed (radius, m s$^{-1}$), and 30 min averaged $N_2O_5$ mole ratios (a) and $ClNO_2$ mole ratios (b) on colorscales. Plots of nighttime $N_2O_5$ (c) and $ClNO_2$ (d) mole ratios vs wind speed, with wind direction shown as a colorscale. No clear correlation was observed between wind speed or direction and $N_2O_5$ or $ClNO_2$ abundance." Section 3.5 now includes a statement of this lack of correlation, as well as the following discussion: "Futher, under the low wind speed conditions of the campaign (nighttime median 1.0 m/s), the gas mole ratios are expected to be higher in response to decreased atmospheric dispersion of gases and emissions/deposition from nearby sources/sinks."

[Figure]

**Reviewer #1**

MAJOR COMMENTS

Most of the analysis presented in this paper uses 30 minutes averaged data. Is this appropriate for this type of study? For example, the periods of fog and snowfall discussed on pages 14-16 are of the order of 1 hour, so maybe higher frequency data would provide more accurate information. I think the authors should comment on this point early on in the paper.

> To clarify this choice, we added the following text in Section 2.1: "*Weather conditions were reported with 1 h resolution. This relatively long time resolution limits the use of higher frequency data from other measurements, and therefore, we use 30 min averaged data, with the assumption that the weather conditions lasted the entire hour.*"

In section 2.2 the authors say that Cl2 and HNO3 were being measured. However these species do not seem to be mentioned in the rest of the paper. If the reason is that they were not observed, I suggest this part of the method is removed. If they were observed, I wonder why they were not used in the subsequent analysis.

> The presence of $Cl_2$ in urban areas is a current research question being discussed (e.g., Liu et al., 2017), and therefore, we believe it is important to state that $Cl_2$ was rarely observed above the 0.6 ppt limit of detection, which we have now clarified in Section 2.2. While $HNO_3$ data is not extensively discussed in the manuscript, we feel it is important to report, given few wintertime, mid-latitude measurements. Therefore, we kept the $HNO_3$ data in Figure S9 and associated discussion in its caption. We also added the following sentence to Section 2.2: "*These upper limits for $Cl_2$ and $HNO_3$ mole ratios are important to report, given limited measurements of these compounds in urban, snow-covered environments.*"

In section 3, it would be good to better define the conditions encountered during the campaign. From figure 1, it seems there was only 1 case of clear sky, 1 of snowfall, 1 of fog and 1 of rain during the entire period, but I suppose these are only selected case studies. What were the conditions on the other days? How were the case studies selected (i.e. are they truly representative of the respective conditions)? Related to this point, it should be clarified how are the statistics in the first part of section 3.1 - including table 1 - calculated: do they refer to the 4 case study nights only, or include other similarly classified periods? This is important to understand how representative are the numbers and how robust is the analysis.

> The previous Figure S2 showed the time periods of snowfall, fog, and rainfall that occurred throughout the entire campaign. To clarify this point and at the Editor's suggestion, we switched Figures 1 and S2 so that this information is readily available in the main text.

> Section 3.1 and Table 1 refer to the data from the entire field campaign, not just the case study nights. We clarified the text in Section 3.1 to emphasize this: "*Campaign-wide average nighttime (18:00-08:00 EST) $N_2O_5$ and $ClNO_2$ mole ratios during clear conditions and each type of weather event are listed in Table 1, with additional data ($PM_{2.5}$ $Cl^-$ and $NO_3^-$, temperature, relative humidity, and friction velocity) provided in Table S1. Section 3.1 discusses these weather events across the entire campaign; example case studies are discussed in Section 3.2.*"

Regarding the criteria for choosing the case studies, we added the following text in Section 3.2: *"Case study nights were chosen to capture sustained (> 7 h) weather events (clear, snowfall, fog, or rainfall). Nights were also selected based on ground cover and friction velocity imilar to the campaign-wide averages during different types of weather events (Figure S3 and Table S1). For example, snowfall was most often coincident with snow-covered ground and lower turbulence; therefore, the selected snowfall case study night also featured snow-covered ground and lower turbulence."*

I find the discussion in section 3.2 (effects of turbulence) a bit lacking, in the sense that it is not immediately clear what the authors think is the effect of turbulence on N2O5 and ClNO2. Sure, high or low turbulence results in higher or lower concentrations, but why? Is it due to deposition, advection or some other physical process?

We revised this paragraph to focus more clearly on the fact that reduced $N_2O_5$ mole ratios are expected under more stable/lower turbulence conditions, due to the near-surface buildup of vehicle $NO_x$ emitted from a nearby roadway, as observed in previous studies. This is because the reaction of $NO_3$ with NO is rapid (k=$2.6 \times 10^{11} \frac{cm^3}{molec \times s}$), such that the lifetime of $NO_3$ ($N_2O_5$ precursor) in the presence of even small quantities of NO is short (Brown et al., 2007; Wang et al., 2006).

Likewise the first part of section 3.4 (page 21) can be a bit expanded: how do all the factors (turbulence, ground conditions, etc...) tie together and relate to the observed values of N2O5 and ClNO2? A short summary at the end of each section would help driving the point home.

To expand on the discussion of $ClNO_2$ and $N_2O_5$ in Section 3.3, we added a discussion to Section 3.5 of how $N_2O_5$ abundance was highest during higher turbulence conditions, independent of ground cover, and such weather events have higher turbulence most frequently. This added text includes: *"Given that statistically higher mole ratios of $N_2O_5$ were observed under higher turbulence conditions, as shown by campaign averages (Fig. 4), we examine how often higher turbulence conditions occurred during each type of weather event. Higher turbulence ($u_* > 0.25$ m s$^{-1}$) was present for 13%, 9%, 19%, and 17% of the time during clear, snowfall, fog, and rainfall conditions, respectively (Fig. S3)."*

As suggested, we added a short summary to the end of Section 3.3: *"In summary, average $N_2O_5$ mole ratios were significantly higher (p<0.05) at six different 30 min time periods, corresponding to 5.9 times higher $N_2O_5$ mole ratios during higher turbulence conditions, in comparison to lower turbulence conditions. The reduced $N_2O_5$ mole ratios observed under lower turbulence conditions are likely due to the short lifetime of $NO_3$ ($N_2O_5$ precursor, R3) when vehicle $NO_x$ is emitted into the stable boundary layer, as observed in previous studies (Brown et al., 2007; Wang et al., 2006). However, average $ClNO_2$ mole ratios were significantly higher (p<0.05) during four different 30 min time periods, corresponding to 6.3 times higher $ClNO_2$ mole ratios during lower turbulence conditions, in comparison to higher turbulence conditions. This points to a likely surface source of $ClNO_2$ upon surface deposition of $N_2O_5$. Therefore, in Section 3.4, we investigated the influence of ground cover (Section 3.4)."*

We also added a short summary to the end of section 3.4: *"To summarize, there were no statistically significant (p<0.05) differences in average $N_2O_5$ mole ratios over the diel period for snow-covered versus bare ground. However, significantly higher (p<0.05)*

*average ClNO₂ mole ratios were observed over snow-covered ground for 11 (of 28) nighttime 30 min time periods, corresponding to 3.5 times higher ClNO₂ mole ratios over snow-covered ground, in comparison to bare ground. During the same field campaign, net positive (production) fluxes of ClNO₂ were measured over snow-covered ground, and field-based chamber experiments showed that ClNO₂ can be produced from the reaction of N₂O₅ on the saline snowpack (McNamara et al., 2021). The observed enhancement of ClNO₂ over snow-covered ground results herein suggests that snowpack ClNO₂ production was a frequent and significant occurrence across the field campaign (e.g. enough to influence the campaign-wide average results). We investigate the effects of other variables (e.g. PM₂.₅ Cl⁻, NO₃⁻, temperature, relative humidity, O₃ concentration, aerosol surface area, and pressure) in Section 3.5."*

MINOR COMMENTS

line 161: do these times correspond to sunset and sunrise?

The following text was added in Section 2: *"At the start of the campaign (January 20), sunrise was at 08:05 local time (eastern standard time, EST), and sunset was at 17:42. At the end of the campaign (February 24), sunrise was at 07:23 EST, and sunset was at 18:27."*

lines 202-203: doesn't this introduce a bias? Isn't it better to exclude these data from analysis?

Excluding these data biases averages low. Therefore, considering data below LOD as $0.5 \times LOD$ is standard practice when calculating averages (McCormick and Karger, 1980).

lines 215-220 (and elsewhere in section 3.1): are the 18:00-8:00 averages discussed here?

Yes. We clarified this by adding the following text: *"Campaign-wide average nighttime (18:00-08:00 local time) N₂O₅ and ClNO₂ mole ratios during clear conditions and each type of weather event are listed in Table 1, with additional data (PM₂.₅ Cl⁻ and NO₃⁻, temperature, relative humidity, and friction velocity) provided in Table S1. Section 3.1 discusses these weather events across the entire campaign; example case studies are discussed in Section 3.2."*

lines 365-367: if there is no significant difference between high and low turbulence, I think it is a bit misleading to say that values are on average a bit higher with high turbulence. More in general, can the authors speculate on why turbulence does not seem to affect N2O5 levels before 2:00?

Thank you for this suggestion. We removed the comparison of the average values and replaced the discussion with the following text: *"Before 02:00 local time, no statistically significant differences were observed in N₂O₅ abundance between higher and lower turbulence conditions, suggesting that titration of NO₃ (N₂O₅ precursor, R3) by NO was not significant during these time periods."*

table 1: maybe add bare/snow ground?

Thank you for this suggestion. Average N₂O₅ or ClNO₂ mole ratios during snow-covered and bare ground are now included in Table 1, as suggested.

table 2: add the expected scavenging coefficient based on solubility? Otherwise -a statements such as line 285 and 318-319 makes little sense.

> To increase the overall clarity of the manuscript and based on feedback from Reviewer #2, all text discussing scavenging/scavenging coefficients (including the above lines and Table 2) have been removed.

**Reviewer #2**

Major comments

(1) One cannot simply compare N2O5 (and ClNO2) abundances with meteorological conditions as presented in this manuscript (rain, snow, fog - Figure 2, lines 215-216; turbulence - Figure 4, line 360) unless the rates of NO3 production, P(NO3) = k4[O3][NO2], the NO3 loss rates to VOCs and NO, temperature and [NO2] (which affect N2O5 concentration via equilibrium K3), and aerosol surface area chloride abundances were of similar magnitude for these events. It is not at all likely that all of these variables were identical. In fact, Table 3 shows that temperatures were very different, indeed.

> We appreciate the complexity of untangling the effects of the multiple variables on $N_2O_5$ and $ClNO_2$ abundances, and we appreciate the points made here by the reviewer. To more quantitatively address the different variables that affect $[NO_3]$, we constructed a box numerical model, which is described in Section S1, with results shown in Figures S14-S16 and discussed in Section 3.5. This box model shows that the variabilities observed in temperature, pressure, $O_3$ mole ratios, and aerosol surface area between the different case studies are insufficient to explain the variability in $N_2O_5$ and $ClNO_2$ mole ratios measured between these case study nights, and points to the importance of the other processes (precipitation, fog, ground cover, and turbulence) discussed in this manuscript.

(2) The calculation of "gas-phase scavenging coefficients" (Table 2) is questionable as one needs to assume the absence of production and transport terms that also affect mole ratios. The examples cited (lines 280-283) are for molecules (SO2 and NH3) that are relatively unreactive and are mainly primary in origin, but that is not the case for ClNO2 and certainly not for N2O5. Furthermore, the analysis is not robust because vastly different values are obtained depending over what time period the scavenging coefficients are calculated. For example, the data in Figure 3d show an increase in ClNO2 mixing ratios during a rain episode - does this imply that the scavenging coefficient would be negative, and the rain is a source of ClNO2?

> We thank the reviewer for making these points. We removed Table 2, which previously contained scavenging coefficients, and all related discussion of scavenging coefficients from the manuscript.

(3) A large portion of the analytical methods, data set and analysis have been presented elsewhere. The authors should avoid unnecessary repetition (e.g., line 128 - section 2.2. N2O5 and ClNO2 measurements using chemical ionization mass spectrometry (CIMS)). Rather than restating everything here, please simply cite the earlier paper(s) where possible, briefly summarize and note deviations from the earlier work.

> There appears to be a misunderstanding here in that the work presented by McNamara et al. (2021) corresponded to $N_2O_5$ and $ClNO_2$ measurements during vertical profile and snow chamber experiments during the same field campaign. The $N_2O_5$ and $ClNO_2$ data and analysis presented in the current manuscript are not published elsewhere, except that the data are available in a public data archive.

> As described below, this work used the same CIMS instrument as in McNamara et al. (2021) but with a different inlet, which is now described in greater detail in Section 2.2, based on the feedback below.

(4) Throughout the manuscript, there are statements such as "N2O5 was fairly stable" and "ClNO2 increasing steadily", which is grammatically incorrect since molecules cannot be referred to in this way, only their abundances. Consider rephrasing to "Mole ratios of N2O5 ..." or "Mixing ratio of ClNO2".

*We thank the reviewer for this attention to detail. We clarified the phrasing throughout, primarily using the phrasing "mole ratios".*

(5) The manuscript would benefit from more data as only one of each snowfall, fog and rainfall events were described in the main paper yet more were observed (Figure S2). It is thus unclear how representative the events selected in the main manuscript are.

*We discuss campaign-wide average mole ratios for each type of weather event (snowfall, fog, and rainfall), in addition to the representative events (case studies). New section headings have been added to try to draw attention to this campaign-wide analysis, and the text has been revised to emphasize this. In addition, the prior Figures S2 and 1 have been switched to more clearly illustrate that snowfall, fog, and rainfall occurred with high frequency throughout the entire field campaign. We also updated the caption for Figure 2 to make it clearer that it considers the data from weather events during the entire campaign, and not just the case studies.*

*We also added the following text to clarify how case study nights were chosen to be as representative as possible relative to the campaign averages during these weather event types, as now described in the following added text: "Case study nights were chosen to capture a sustained weather event (e.g. >7 h of clear conditions, snowfall, fog, or rainfall). Additionally, ground cover and friction velocity were matched as closely as possible for case study nights to the campaign-wide averages during different types of weather events."*

Specific comments

lines 153-54. "N2O5 and ClNO2 were calibrated offline relative to Cl2 as described in McNamara et al. (2019b)." A better way to say this is "the instrument response for N2O5 and ClNO2 was calibrated ..." as a compound cannot be calibrated.

*To address this, the text now reads: "The instrument responses for $N_2O_5$ and $ClNO_2$ were calibrated in the laboratory, with calibration factors relative to the response to $Cl_2$ obtained, as described by McNamara et al. (2019)."*

McNamara et al. (2019b) did not describe a calibration "relative to Cl2" which would not be accurate since Cl2 does not convert quantitatively to ClNO2 (and cannot be used to calibrate for N2O5); instead, they described a titration method for N2O5 and thermal dissociation method for ClNO2.

Please clarify how response factors were obtained in this work and state how accurate the derived calibration factors were.

*As described in McNamara et al. (2019), ClNO$_2$ was calibrated in the laboratory using the method of Thaler et al. (2011), and N$_2$O$_5$ was calibrated using the method of Bertram et al. (2009). By "relative to Cl$_2$", we mean that a calibration scaling factor was obtained. In-field CIMS calibrations were conducted every 2 h for Cl$_2$. Then in the laboratory, ClNO$_2$ calibration was performed, and the Cl$_2$ response from the Cl$_2$ permeation source was simultaneously determined to calculate a calibration scaling factor (ClNO$_2$ response/Cl$_2$*

response) to allow the lab-determined ClNO$_2$ calibration factor to be applied to the field data. Uncertainties in the relative sensitivity factors come from variations in the Cl$_2$ permeation source output during in-field calibrations, variations in the flow rate pulled by the CIMS, and fluctuations in the normalized CIMS signals for both Cl$_2$ and the compound being calibrated relative to Cl$_2$. The relative sensitivity factors (relative to Cl$_2$) ± uncertainty were 3.7±0.4 hz hz$^{-1}$ ppt$^{-1}$ and 0.85±0.08 hz hz$^{-1}$ ppt$^{-1}$ for ClNO$_2$ and N$_2$O$_5$, respectively. This is a common method applied to CIMS data (e.g., Liao et al., 2011; McNamara et al., 2019) when it is not feasible to calibrate all chemical species regularly in the field and to account for any response differences (e.g., due to changing detector sensitivity) between the field and lab. We now clarify this in Section 2.2.

N2O5 is not quantitatively transmitted through inlets. What was assumed for its inlet transmission efficiency? How uncertain and variable is the inlet transmission efficiency?

In Section 2.2, we now describe and clarify that the inlet used is a high flow inlet commonly used for measurements of radicals (Liao et al., 2011). This is in contrast to the longer inlet used for vertical profile measurements during the SNACK campaign by McNamara et al. McNamara et al. (2021). Added text includes:  *"The CIMS was housed in a mobile laboratory trailer at the field site, and sampled ambient air at ~300 L min$^{-1}$ through a specialized inlet, designed to prevent wall losses of reactive species by allowing for the sampled air at the center of the ring to be de-coupled from the inlet walls (laminar flow), thereby avoiding wall surfaces* (Huey et al., 2004; Neuman et al., 2002)*, as in previous campaigns (e.g.,* McNamara et al. (2019))*."* This inlet has previously been shown by Tanner and Eisele (1995) to have <10% losses for the hydroxyl radical and to result in [BrO] within measurement uncertainty of differential optical absorption spectroscopy measurements (Liao et al., 2011). Here we assume that any minor loss observed would be within the stated 22%+0.3 ppt uncertainty for 30 min averaged N$_2$O$_5$ mole ratios.

line 164. "Cl2 was monitored" appears under the heading "N2O5 and ClNO2 measurements ..." Please address this (minor) organizational issue.

The title of the heading for section 2.2 has been updated to: "*Chemical ionization mass spectrometry measurements*".

line 226 Table 1, caption "±95% confidence interval". These are very small confidence intervals, too small to be credible in my opinion. How were the CI calculated? Do the values in the table take uncertainties in calibration factors and inlet transmission factors into account, or are they were merely calculated based on (averaged?) measurement precision?

Confidence intervals (CI) were calculated using the standard definition:

$$CI = z \times \frac{\sigma}{\sqrt{n}}$$

where z=the z-value (z=1.96 for the 95% confidence interval), σ=standard deviation, and n=number of data points. In Table 1, these confidence intervals are meant to represent the variabilities in the observations under the different weather conditions. Therefore, the following sentence was added to the Table 1 caption: "*95% confidence intervals are reported to describe the variabilities in 30 min averaged values of the parameters for the various weather and ground cover conditions.*" The propagated measurement uncertainties in N$_2$O$_5$ and ClNO$_2$ are reported in Section 2.2 to be 22%+0.3 ppt and 22%+0.1 ppt for 30 min averaged N$_2$O$_5$ and ClNO$_2$ mole ratios, respectively.

line 253 / Figure 3. One of the axis titles is missing an oxygen.

We thank the reviewer for identifying the omission of oxygen in the axis title. We fixed the erroneous axis title.

For the snowfall case (Figure 3b) there appears to be a sustained loss throughout the episode with a consistent loss rate coefficient, but for the other cases (fog - 3c and rain 3d) the mole ratios sometimes increase during the episode. Please explain why this might be and how this affects the subsequent analysis.

To improve the overall clarity of the manuscript, we removed the calculated scavenging coefficients and all related discussion. Now the case studies used to convey the trends in $N_2O_5$ and $ClNO_2$ more generally, and to illustrate that they are lower during rainfall and fog, in comparison to clear weather conditions. In section 3.2, we now also offer an explanation for mole ratios sometimes briefly increasing during episodes of snowfall, rainfall, or fog: "*Variations in N2O5 and ClNO2 over the course of the nights are likely due to variability in fog/rainfall that were not resolved by the time resolution of the reported weather conditions, which also do not reflect precipitation rates or fog concentrations that would be expected to vary through the nights.*"

line 280-281. "one hour fog period". Where all scavenging coefficients calculated over 1-hr long periods? How did the authors decide over what periods the loss rates should be calculated?

Table 2 and all discussion of scavenging coefficients have been removed.

line 289 / Table 2. Uncertainty estimates should be added to Table 2. Please indicate (in Figure 3) over what periods the scavenging coefficients were calculated, as the derived values depend on it.

Table 2 and all discussion of scavenging coefficients have been removed.

line 300. "Although precipitation rates were used to inform time periods used for calculations during the rainfall case, a more thorough characterization of scavenging with respect to precipitation rate and intensity is beyond the scope of this discussion." This would have been useful, imo.

Table 2 and all discussion of scavenging coefficients have been removed.

line 360. Here, N2O5 mole ratios are compared to turbulence conditions. However, this analysis is not sound as it is not clear if the production from oxidation (via reaction of NO2 with O3 to NO3 and subsequent reaction with NO2), sinks (e.g., aerosol surface area, VOC abundance) and temperature (which shifts the NO2/NO3/N2O5 equilibrium and has a large effect on N2O5 concentration and loss rates) are identical for the high and low turbulence cases.

As stated above, we constructed a simple box model, which is described in Section S1, with results shown in Figures S14-S16 and discussed in Section 3.5. This box model shows that the variabilities observed in temperature, pressure, $O_3$ mole ratios, and aerosol surface area between the different case studies (clear, snowfall, fog, rainfall) are insufficient to explain the variability in $N_2O_5$ mole ratios observed between these case study nights, and points to the importance of the other processes (precipitation, fog, ground cover, and turbulence) discussed in this manuscript.

To clarify the discussion of turbulence effects on $N_2O_5$ in section 3.3, we have removed some information that was less relevant and now focus this paragraph on the likely role of nearby roadway $NO_x$ emissions, which we expect to control the $NO_3$ titration and limited $N_2O_5$ levels observed under lower turbulence conditions, as observed in other studies (Brown et al., 2007; Wang et al., 2006).

line 464. The N2O5 mixing ratios observed were small; how much additional nitrate would be expected if all of it were taken up (i.e., if the production of N2O5 via NO3, i.e., R4, were integrated)?

Thank you for this suggestion. The following text in Section 3.5 now states: "*On average, $N_2O_5$ was 76±5 ppt lower during fog compared to clear conditions (**Figs. 2** and **S10**). While this difference is not completely attributable to $N_2O_5$ uptake, it would correspond to a nitrate concentration of 0.21 µg m$^{-3}$.*" For context, $PM_{2.5}$ $NO_3^-$ concentrations were higher by 0.43±0.06 µg m$^{-3}$, as also stated in that section, with $HNO_3$ uptake also attributed to the increased nitrate concentration during fog."

line 565 - strike "Received"

"Received" has been removed.

Figure S1 - please state uncertainty of the slope. What is the theoretical value based on? Note that the 37Cl :35Cl isotope ratio is known to higher precision than shown.

The value ±RMSE for the slope is 0.316±0.005. The theoretical value is 0.31996. Both values have been updated in Fig. S1.

**Reviewer #3**

Major comments

Why did the authors choose to use 30-minute averages to analyze their data? It seems that faster data would be useful for this type of analysis. Please explain.

*To clarify this choice, we added the following text in Section 2.1: "Weather conditions were reported with a maximum time resolution of 1 h. This relatively long time resolution limits the use of higher frequency data from other measurements, and therefore, we use 30 min averaged data, with the assumption that the weather condition lasted the entire hour."*

Generally, there is a lack of information about what criteria were used to classify the different conditions. This should be added.

*In Section 2.1, we added the following description: "Weather conditions were classified using reported National Weather Service designations: clear weather conditions include fair, cloudy, mostly cloudy, and partly cloudy; snowfall includes light snow, snow, heavy snow, and wintry mix; fog includes fog and haze; and rainfall refers to light rain, rain, heavy rain, and thunderstorms."*

There is also a lack of information about the number of samples used to assess each condition. For example, in Section 3.1, mixing ratios for each type of weather condition are compared graphically and statistically, but the reader has no indication of how many events or 30-minute time points were considered for each condition. It would be useful to include this information throughout the manuscript: the overall clear/rain/snow/fog conditions (e.g. in Table 1), the low/high turbulence conditions (e.g. in caption of Figure 4), ground cover (e.g. in caption of Figure 5).

*Thank you for this helpful suggestion to improve clarity. Figure S2 was moved to the main text (now Figure 1), and this visually shows the time periods of the various weather conditions. We also made the additions to Table 1 and the captions of Figures 4 and 5 as suggested.*

As I was reading Section 3.1, several questions arose about the impacts of meteorological conditions, many of which were addressed in Section 3.4. To reduce confusion, I suggest adding text to Section 3.1 indicating that the effects of RH and T will be discussed later.

*As suggested, the following text was added: "The effects of temperature and relative humidity are discussed in Section 3.5."*

It would also help the reader to combine Tables 1 and 3 as I found myself flipping back and forth between the two. One of my questions in Section 3.1 that was not answered in the manuscript was the impact of wind direction (if any) on the observed differences in N2O5 and ClNO2 under different weather conditions. This should be added to the manuscript.

*Tables 1 and 3 were combined as suggested.*

*As is shown in the newly added Figure S13 (included in the response to the Editor), no correlation was observed between wind speed or direction and $N_2O_5$ or $ClNO_2$ abundances. This is also now discussed in Section 3.5.*

Specific comments:
Line 69: Define NOx at first usage (line 39)

$NO_x$ is now defined as $NO+NO_2$.

Line 73: Should be equilibrium arrows (can be inserted in Word by typing 21cc, then pressing ALT and "x" simultaneously)

We made this change, and thank the reviewer for giving such detailed information about how to insert the equilibrium arrows!

Line 187: Where was LiF added? Was it an internal standard for chromatography and/or particle sampling?

LiF was added as an internal standard to the hydrogen peroxide solution that supplies the parallel plate wet denuder, as described in Markovic et al. (2012) and used as an indicator of a change in capacity of the concentrator columns for the gas-phase channels.

Figure 3: Check y-axis label in Figure 3a

We thank the reviewer for their attention to detail. We fixed the erroneous axis title.

Table 2: Present the scavenging coefficients in the table in the same order as they are discussed in the text.

Table 2 and all discussion of scavenging coefficients have been removed based on feedback from Reviewer #2.

Line 265: Is this different from the dimensionless Henry's Law coefficient (or air-water partitioning coefficient, K(AW))?

The following text has been added to improve clarity: *"Converting the $K_H$ for $ClNO_2$ to its dimensionless Henry solubility (also called the air-water partitioning coefficient, $K_{AW}$), as in* Sander (2015)*, gives a unitless ratio between the aqueous and gas phases of >1 at temperatures above freezing."*

Line 330: Would be clearer to define the trend (i.e. thickens with increasing temperature). Listing the average temperature during the snowfall case in the text here would help to clarify.

These sentences discussing scavenging/scavenging coefficients have been removed, based on feedback from Reviewer #2.

References: McNamara et al. 2020a and Sander et al. 2015 are missing from the reference list.

McNamara et al. 2020a was corrected to simply McNamara et al. 2020. Sander (2015) is now included in the reference list."

**References:**

Bertram, T. H., Thornton, J. A. and Riedel, T. P.: Atmospheric Measurement Techniques An experimental technique for the direct measurement of N2O5 reactivity on ambient particles. [online] Available from: www.atmos-meas-tech.net/2/231/2009/ (Accessed 6 May 2019), 2009.

Brown, S. S., Dubé, W. P., Osthoff, H. D., Stutz, J., Ryerson, T. B., Wollny, A. G., Brock, C. A., Warneke, C., de Gouw, J. A., Atlas, E., Neuman, J. A., Holloway, J. S., Lerner, B. M., Williams, E. J., Kuster, W. C., Goldan, P. D., Angevine, W. M., Trainer, M., Fehsenfeld, F. C. and Ravishankara, A. R.: Vertical profiles in $NO_3$ and $N_2O_5$ measured from an aircraft: Results from the NOAA P-3 and surface platforms during the New England Air Quality Study 2004, J. Geophys. Res. Atmos., 112(22), 1–17, doi:10.1029/2007JD008883, 2007.

Huey, L. G., Tanner, D. J., Slusher, D. L., Dibb, J. E., Arimoto, R., Chen, G., Davis, D., Buhr, M. P., Nowak, J. B., Mauldin, R. L., Eisele, F. L. and Kosciuch, E.: CIMS measurements of $HNO_3$ and $SO_2$ at the South Pole during ISCAT 2000, Atmos. Environ., 38, 5411–5421, doi:10.1016/j.atmosenv.2004.04.037, 2004.

Liao, J., Sihler, H., Huey, L. G., Neuman, J. A., Tanner, D. J., Friess, U., Platt, U., Flocke, F. M., Orlando, J. J., Shepson, P. B., Beine, H. J., Weinheimer, A. J., Sjostedt, S. J., Nowak, J. B., Knapp, D. J., Staebler, R. M., Zheng, W., Sander, R., Hall, S. R. and Ullmann, K.: A comparison of Arctic BrO measurements by chemical ionization mass spectrometry and long path-differential optical absorption spectroscopy, J. Geophys. Res. Atmos., 116, 1–14, doi:10.1029/2010JD014788, 2011.

Liu, X., Qu, H., Gregory Huey, L., Wang, Y., Sjostedt, S., Zeng, L., Lu, K., Wu, Y., Hu, M., Shao, M., Zhu, T. and Zhang, Y.: High Levels of Daytime Molecular Chlorine and Nitryl Chloride at a Rural Site on the North China Plain, Enviornmental Sci. Technol., 51, 9588–9595, doi:10.1021/acs.est.7b03039, 2017.

Markovic, M. Z., Vandenboer, T. C. and Murphy, J. G.: Characterization and optimization of an online system for the simultaneous measurement of atmospheric water-soluble constituents in the gas and particle phases, J. Environ. Monit., 14, 1872–1884, doi:10.1039/c2em00004k, 2012.

McCormick, R. M. and Karger, B. L.: Guidelines For Data Acquisition And Data Quality Evaluation In Environmental Chemistry, Anal. Chem., 52(14), 2242–2249, doi:10.1021/ac50064a004, 1980.

McNamara, S. M., W. Raso, A. R., Wang, S., Thanekar, S., Boone, E. J., Kolesar, K. R., Peterson, P. K., Simpson, W. R., Fuentes, J. D., Shepson, P. B. and Pratt, K. A.: Springtime Nitrogen Oxide-Influenced Chlorine Chemistry in the Coastal Arctic, Environ. Sci. Technol., 53, 8057–8067, doi:10.1021/acs.est.9b01797, 2019.

McNamara, S. M., Chen, Q., Edebeli, J., Kulju, K. D., Mumpfield, J., Fuentes, J. D., Bertman, S. B. and Pratt, K. A.: Observation of N2O5 deposition and ClNO2 production on the saline snowpack, ACS Earth Sp. Chem., doi:10.1021/acsearthspacechem.0c00317, 2021.

Neuman, J. A., Huey, L. G., Dissly, R. W., Fehsenfeld, F. C., Flocke, F., Holecek, J. C., Holloway, J. S., Hübler, G., Jakoubek, R., Nicks, D. K., Parrish, D. D., Ryerson, T. B., Sueper, D. T. and Weinheimer, A. J.: Fast-response airborne in situ measurements of $HNO_3$ during the Texas 2000 Air Quality Study, J. Geophys. Res. D Atmos., 107, 1–12, doi:10.1029/2001JD001437, 2002.

Sander, R.: Compilation of Henry's law constants (version 4.0) for water as solvent, Atmos. Chem. Phys, 15, 4399–4981, doi:10.5194/acp-15-4399-2015, 2015.

Tanner, D. J. and Eisele, F. L.: Present OH measurement limits and associated uncertainties, , 100(94), 2883–2892, doi:10.1029/94JD02609, 1995.

Thaler, R. D., Mielke, L. H. and Osthoff, H. D.: Quantification of nitryl chloride at part per trillion mixing ratios by thermal dissociation cavity ring-down spectroscopy, Anal. Chem.,

83(7), 2761–2766, doi:10.1021/ac200055z, 2011.

Wang, S., Ackermann, R. and Stutz, J.: Vertical profiles of $O_3$ and $NO_x$ chemistry in the polluted nocturnal boundary layer in Phoenix, AZ: I. Field observations by long-path DOAS, Atmos. Chem. Phys., 6(9), 2671–2693, doi:10.5194/acp-6-2671-2006, 2006.

---

## Author Response (AR2)

Thank you to the editor for their review and the opportunity to make the corrections noted below. Our response is provided in blue.

**Comments to the author**:
The authors have done a good job responding to the comments and suggestions in the original round of reviews. The box model analysis provides solid indirect evidence of the importance of scavenging processes in that omitting them causes large biases. This approach is an improvement over the scavenging analysis in the original manuscript and addresses reviewers' concerns about that section.

I find the paper to now be ready for final publication in ACP

Two small technical issues to address:
Lines 65 and 510-511 – suggest slight rewording to clarify that higher N2O5 abundance or net production is favoured at low temperature (i.e., the production rate may not actually be faster since NO2 + O3 has a high energy barrier)

> As recommended, Lines 65-66 now state: "The net formation of $N_2O_5$ from $NO_2$ and $NO_3$ is a temperature-dependent equilibrium, with net $N_2O_5$ production favoured at lower temperatures…"

> Lines 510-511 now state "Since net $N_2O_5$ production (**R3**) is favoured at lower temperatures…", as recommended.

Line 482 – add a comma after "clear"

> This was fixed.